# GPerturb: Gaussian process modelling of single-cell perturbation data

Hanwen Xing[1] & Christopher Yau [1,2] ✉

Single-cell RNA sequencing and CRISPR screening enable high-throughput analysis of genetic perturbations at single-cell resolution. Understanding combinatorial perturbation effects is essential but challenging due to data sparsity and complex biological mechanisms. We present GPerturb, a Gaussian process-based sparse perturbation regression model designed to estimate gene-level perturbation effects. GPerturb employs an additive structure to separate signal from noise and captures sparse, interpretable effects from both discrete and continuous responses. It also provides uncertainty estimates for the presence and strength of perturbation effects on individual genes. We demonstrate the use GPerturb on both simulated and real-world datasets, characterising its competitive performance with current state-of-the-art methods. Furthermore, the model reveals meaningful gene-perturbation interactions and identifies effects consistent with known biology. GPerturb offers a novel approach for uncovering complex dependencies between gene expression and perturbations and advancing our understanding of gene regulation at the single-cell level.

Developments in Single-cell RNA sequencing (scRNA-seq) and Clustered Regularly Interspaced Short Palindromic Repeats (CRISPR) screening accelerate the discovery of association between genes and various biological processes such as immune responses, cell proliferation and drug resistance[1–6]. In particular, technologies such as CROP sequencing (CROP-seq)[7] and Perturb sequencing[1] (Perturb-seq) have made high-throughput, large scale cellular perturbation screens possible. Such cellular perturbation screens allow practitioners to investigate complex biological mechanisms such as regulatory dependencies and drug responses on a single-cell level using the comprehensive, fine-grained readouts of the target perturbations within single cells, and have found applications in studies such as combinatorial therapy[8,9], drug discovery[10] and regulatory elements[11,12].

The growing granularity of measurements provided by single-cell CRISPR screening technologies motivates the need for novel computational methods to help extracting interpretable biological insights from generated data particularly in relation to the perturbation effects. However, it is a challenging task due to the high dimensionality, complex structure and sparse nature of the single-cell screening measurements. Analytically, the problem is to produce a prediction model that can be used to provide an estimate of the effect of a perturbation on expression for any particular cell type or cell. The model can be developed using a training dataset that consists of a series of expression measurements on single cells, in which each cell belongs to one of a finite number of cell types and has been subject to one of a finite number of perturbations (including unperturbed controls).

A common approach has been to apply deep learning-based techniques to learn the relationships between cell type, perturbation and expression output flexibly from sufficiently large datasets. To do this, expression data and cell type information are typically transformed into *embeddings* via deep neural networks (DNNs), which are learnt low-dimensional projections of the original measurements, and then the effects of the perturbations are described in this embedding space. Perturbed embeddings can then be remapped back to expression measurements. If there are a large number of perturbations, embeddings may also be formed for the perturbations themselves. Models are trained against an objective which seeks to minimise the discrepancy between the observed perturbation effect and that

[1]Nuffield Department for Women's and Reproductive Health, University of Oxford, Oxford, UK. [2]Health Data Research UK, London, UK.
✉ e-mail: christopher.yau@wrh.ox.ac.uk

predicted via the model. A bottlenecking effect in the design of the model architectures, which perform the various embedding and output transformations, leads to the creation of compressed representations that are optimised towards the maximal retention of information.

The Compositional Perturbation Autoencoder[13] (CPA) is an example of such an approach. Given the measured unperturbed and perturbed expression of a cell, CPA predicts the counterfactual distribution of the expression of that cell had it been subjected to a different perturbation. CPA adopts an autoencoder learning framework and uses additive latent embedding of the cell and perturbation states. SAMS-VAE[14] using a sparse additive mechanism shift variational autoencoder to characterise perturbation effects as sparse latent representations. In SAMS-VAE, the latent representation of a perturbed expression vector is obtained by adding a sparse representation of the perturbation to a dense perturbation-independent basal state, and the decoder is trained to reconstruct the perturbed expression vectors from latent representations. Other approaches have sought to embed additional external information about the expression features to improve predictions. GEARS[15] uses a knowledge graph of gene-gene relationships to inform the prediction, allowing it to simulate the outcomes of perturbing unseen single genes or combinations of genes. Although not deep learning-based, CellOT[16] leverages DNNs for function estimation in a neural optimal transport framework[17] to map between unperturbed and perturbed single-cell responses.

More recently, single cell foundation models[18–21] have emerged, which promise to provide a multi-functional basis for many analytical applications[22]. However, the benefits of current foundation model approaches are not yet clear[23–26] and fair evaluation is complicated by emerging applications that integrate direct empirical data with knowledge extracted from scientific literature or pre-trained foundation models[25,27].

Non-deep learning approaches have also been developed. Guided Sparse Factor Analysis[28] (GSFA) models continuous observations and adopts a linearity assumption in its multivariate latent factor regression approach.[12] uses a variant of the "factorize-recover" algorithm to infer perturbation effects from composite sample phenotypes from compressed Perturb-seq experimental data using combination of sparse principal components analysis and LASSO regression. The attractiveness of such approaches is their comparative simplicity due to the use of linearity assumptions.

The plethora of computational perturbation modelling methods[29,30] disguises many practical issues that are only apparent at usage time. For instance, assumptions in experimental setup and data preprocessing can be implicitly built into models. CPA assumes categorical cell-level information and continuous gene expression inputs but SAMS-VAE is not able to incorporate additional cell-level information, such as batch information or cell type, and can only handle binary perturbation and count-based expression inputs. GSFA utilises its own particular approach to input data transformation and normalisation. While CPA is able to process continuous perturbation levels (e.g., dosage), GEARS applies only to discrete perturbations as it uses perturbation embeddings and relational graphs between perturbations. Foundation models often require their own specific approaches for tokenisation and input data embedding. These model design differences, on a practical level, mean direct and intuitive comparisons between methods may not be possible both in terms of their predictions but also in terms of the explanations underlying those.

In this work, we propose a more conceptually classical approach for perturbation modelling, called **GPerturb**, which utilises hierarchical Bayesian modelling[31] and Gaussian Process regression[32]. We demonstrate that **GPerturb** can achieve high levels of predictive performance that is comparable to current state of the art perturbation models even using a sparse, additive modelling structure and without the use of latent embeddings or external information. Further, the modularity of the hierarchical construction allows us to examine the effect of swapping an observational data model based on count-based expression data for one which uses continuous transformed values instead. Despite the abundance of perturbation modelling methods available, **GPerturb** offers a novel and scalable generative modelling approach with classical features which make prediction output and their interpretation more readily understandable compared to methods based on black box learning.

## Results

### Overview

We first provide an overview of **GPerturb** (a more detailed mathematical description is provided in "**Methods**"), which is a generative model that aims to directly identify and estimate sparse, interpretable gene-level perturbation effects, for analysing single-cell CRISPR screening data. In **GPerturb**, we assume that each expression feature measured for each cell can be explained as a sample from a distribution. In the case where expression data is continuous, a normal distribution is used (zero-inflated Poisson for count-based data), where the mean expression level is given by the combination of two components. The first is a feature-specific basal expression level which is determined by the cell-specific parameters (e.g., cell type or cell-specific sequencing information). The second component is a feature-specific perturbation effect which depends on the type of perturbation applied to the cell (which can be null). To make it explicit that some perturbations will only affect certain features, the perturbation component for each feature is controlled by a binary on/off switch. The relationships that map cell-specific parameters and perturbation type to the observed expression levels are governed by nonlinear Gaussian processes.

**GPerturb** adopts a supervised learning approach to learn and disentangle the basal (unperturbed) expression distribution associated with a given cell type and the additive effect of perturbations given observed gene expression measurements (Fig. 1A). Gaussian processes[32] are used to model expression functions, and sparsity constraints aim to regularise the model and improve generalisation and robustness. The generative properties of **GPerturb** allow perturbed expression levels to be predicted (Fig. 1B), and the sparsity in perturbation effects facilitate users in identifying details about complex perturbation-gene dependencies.

Compared with existing methods, **GPerturb** does not require a latent variable construction and incorporates uncertainty propagation in an intuitive way due to the Bayesian framework. It can be applied to either raw count (**GPerturb**-ZIP, for zero-inflated Poisson) or continuous transformed expression measurements (**GPerturb**-Gaussian). Further and detailed information about the model development and relationships to existing methods can be found in "**Methods**".

### Single-gene perturbation analysis

We first compared the predictive performance of GPerturb, CPA, GEARS and SAMS-VAE on a subset of the genome-wide CRISPR interference Perturb-seq dataset.[33] For all methods, the recommended settings are used. Since SAMS-VAE takes count-based data as inputs while CPA and GEARS require continuous expression inputs, we compare SAMS-VAE against **GPerturb**-ZIP and CPA and GEARS against **GPerturb**-Gaussian, respectively. Similar to previous studies, we randomly select 20% of the dataset as the test set, and use the rest to train GPerturb.

We compared the *averaged* predictions and *averaged* observations for each of the unique perturbations. For CPA, SAMS-VAE and GEARS, we computed and store the *average* of 1000 samples of reconstructed/predicted expressions drawn from the fitted model for each of the unique perturbations. Similarly, for **GPerturb**, we compute and store the *averaged* predicted mean expressions for each of the unique perturbations (i.e., averaged over all samples associated with a common perturbation). Table 1 shows the Pearson correlations

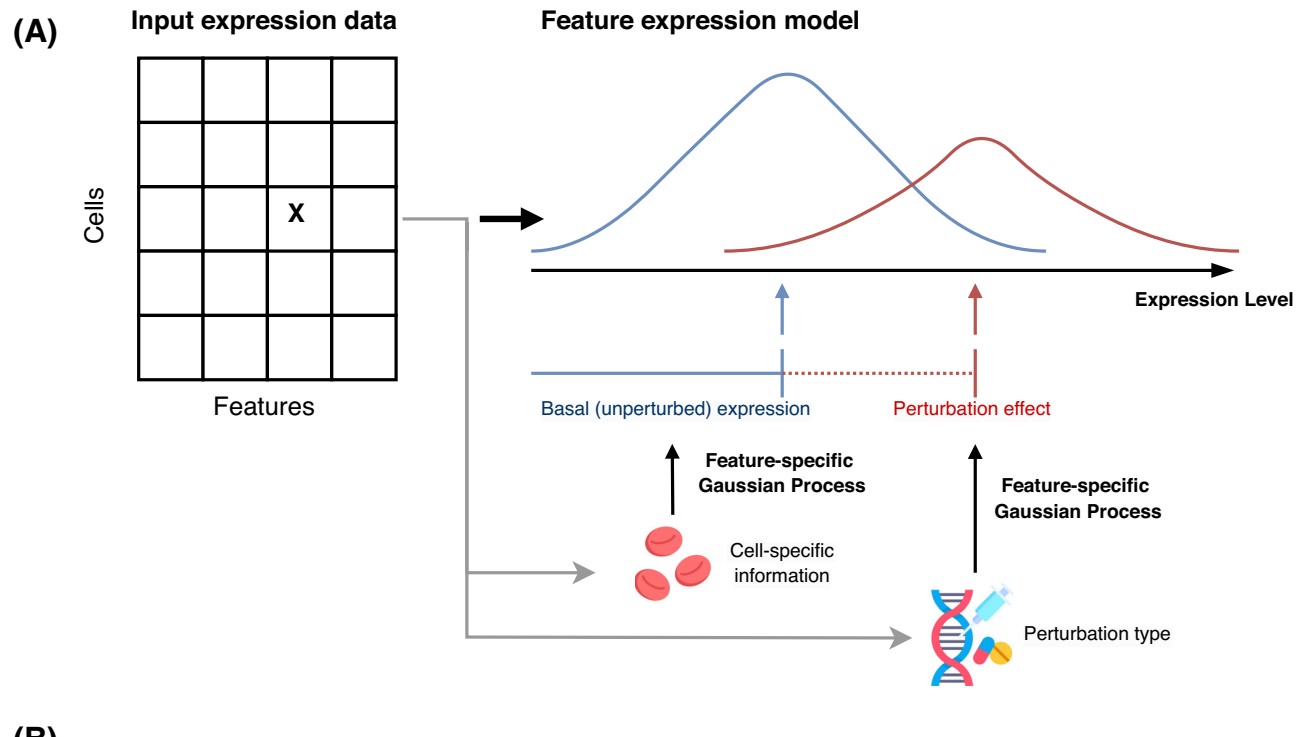

**(A)**

Input expression data

Feature expression model

Cells / Features / X

Expression Level

Basal (unperturbed) expression

Perturbation effect

Feature-specific Gaussian Process

Cell-specific information

Feature-specific Gaussian Process

Perturbation type

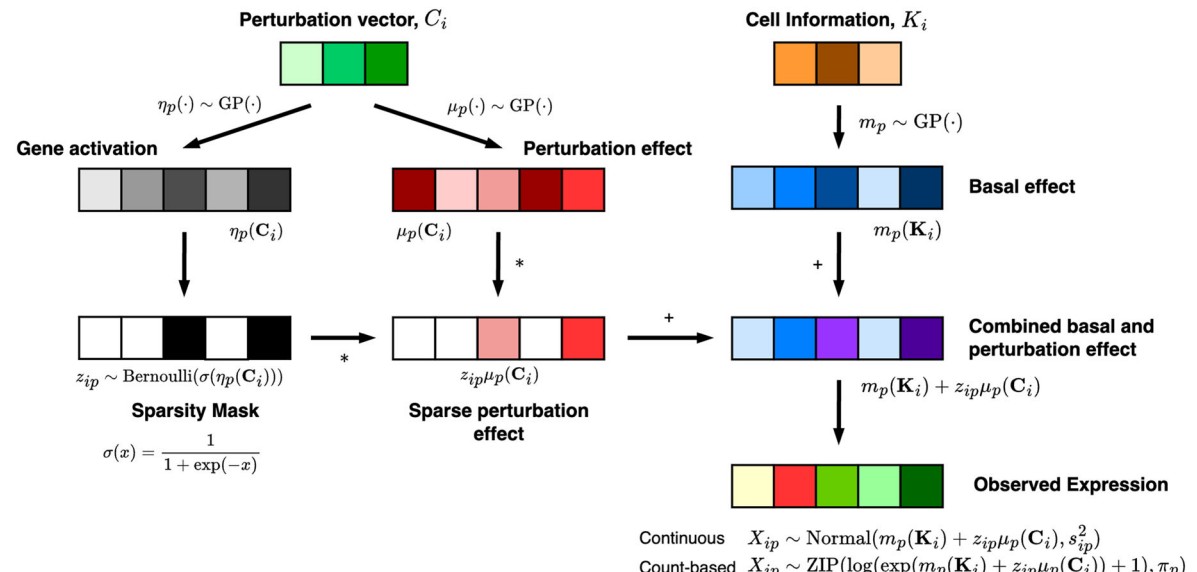

**(B)**

Perturbation vector, $C_i$

Cell Information, $K_i$

$\eta_p(\cdot) \sim \mathrm{GP}(\cdot)$     $\mu_p(\cdot) \sim \mathrm{GP}(\cdot)$     $m_p \sim \mathrm{GP}(\cdot)$

**Gene activation**          **Perturbation effect**          **Basal effect**

$\eta_p(\mathbf{C}_i)$          $\mu_p(\mathbf{C}_i)$          $m_p(\mathbf{K}_i)$

$z_{ip} \sim \mathrm{Bernoulli}(\sigma(\eta_p(\mathbf{C}_i)))$     $z_{ip}\mu_p(\mathbf{C}_i)$     **Combined basal and perturbation effect**

**Sparsity Mask**          **Sparse perturbation effect**          $m_p(\mathbf{K}_i) + z_{ip}\mu_p(\mathbf{C}_i)$

$\sigma(x) = \dfrac{1}{1 + \exp(-x)}$

**Observed Expression**

Continuous     $X_{ip} \sim \mathrm{Normal}(m_p(\mathbf{K}_i) + z_{ip}\mu_p(\mathbf{C}_i), s_{ip}^2)$
Count-based     $X_{ip} \sim \mathrm{ZIP}(\log(\exp(m_p(\mathbf{K}_i) + z_{ip}\mu_p(\mathbf{C}_i)) + 1), \pi_p)$

**Fig. 1 | Overview of GPerturb. A** For each cell-feature, **GPerturb** models the distribution over observed feature expression as the sum of a basal (unperturbed) expression and perturbation effect using feature-specific Gaussian Processes to transform of cell-specific information and perturbation applied. **B** Schematic of the mathematical construction of **GPerturb** showing the incorporation sparsity models and the ability to provide observation models for continuous and count-based expression data. The red blood cells and gene therapy icons: Flaticon.com. This cover has been designed using resources from Flaticon.com.

between the predicted and observed expression levels for the perturbations which are illustrated in Fig. 2A, C. We see **GPerturb**-ZIP attains better correlation than SAMS-VAE ($r_{\mathrm{GPerturb}} = 0.972$, $r_{\mathrm{SAMS\text{-}VAE}} = 0.944$) for count-based inputs, while CPA-mlp achieved the best performance ahead of **GPerturb**-Gaussian and GEARS on continuous inputs ($r_{\mathrm{CPA\text{-}mlp}} = 0.984$, $r_{\mathrm{GPerturb}} = 0.981$, $r_{\mathrm{GEARS}} = 0.977$).

While the overall correlation between predicted and observed expression was high, Fig. 2B, D shows that the directionality of the perturbation effects given by different models did not always agree, with instances where one method might report that a perturbation gives increased gene expression while another method indicates that

the perturbation leads to decreased expression. We quantified this observation in Table 2 by examining the directionality agreement over all gene-perturbation pairs. Figure 3A shows the discrepancies in the exosome-related perturbation effects between **GPerturb**-Gaussian, CPA and GEARS for continuous expression input. In contrast, using count-based expression input, **GPerturb**-ZIP and SAMS-VAE showed greater consistency suggesting that the choice of pre-processing could have a considerable impact on perturbation modelling (Fig. 3B). In order to further examine this, we were able to compare the outputs of **GPerturb**-Gaussian and **GPerturb**-ZIP on 345 perturbations grouped by pathways (Fig. 4). This showed that given the same data set, the

conversion from count-based to continuous-based expression (and the necessary changes in likelihood model in **GPerturb**) considerably changes the predicted perturbation effects.

## Multi-gene perturbation analysis

We next considered a Perturb-seq dataset[34] consisting of 89,357 cells and 5045 genes and containing 131 two-gene perturbations. We compute the averaged predicted responses each method for each of the

two-gene perturbations and compared to the corresponding averaged observations which are shown in Table 1 and Fig. 5A, C. Note that unlike the other methods, GEARS is able to predict perturbation outcomes of previously unseen multi-gene perturbations by using biological knowledge encoded in its knowledge graph. Although **GPerturb** does not use additional prior information as in GEARS, it attains comparable correlation on predictions for two-gene perturbations and outperforms CPA and SAMS-VAE (Table 1). Interestingly, as in the previous experiments, the directionality of the perturbation effects between methods was not always consistent as illustrated in Fig. 5B, D and quantified in Table 2.

We further compare GEARS and **GPerturb** using a highly multiplexed Perturb-seq dataset[12] under the same setup. CPA and SAMS-VAE could not be applied to this data set due to the large number perturbations. We report the averaged predictions against the corresponding averaged observations for both methods in Fig. 5E. We see **GPerturb** and GEARS attain comparable predictions ($r_{\text{GPerturb}} = 0.798$, $r_{\text{GEARS}} = 0.802$), but predictions had high variance. Consequently, estimated perturbation effects for each perturbation-gene pair given by **GPerturb** and GEARS in Fig. 5F showed weak correlation only. Unlike the previous examples, we see that even though the two methods have similar prediction accuracy, the scale of estimated perturbation effects given by GEARS is much smaller than **GPerturb**. The much more conservative perturbation estimations given by GEARS are likely due to the fact that less than 30% of the genetic perturbations is present in the Gene-Ontology (GO) knowledge graph in the current implementation of GEARS.[12]

### Table 1 | Comparison of predictive performance

| EXPRESSION INPUT | METHOD | DATASET | | | |
|---|---|---|---|---|---|
| | | Sciplex2 | Replogle | Norman | Yao |
| Continuous, transformed | GPerturb-Gaussian | **0.988** | 0.981 | 0.979 | 0.798 |
| | CPA-logsig | 0.980 | – | – | – |
| | CPA-mlp | 0.985 | **0.984** | 0.977 | – |
| | GEARS | – | 0.977 | **0.982** | **0.802** |
| Count-based | GPerturb-ZIP | **0.973** | **0.972** | **0.970** | **0.861** |
| | SAMS-VAE | – | 0.944 | 0.969 | – |

Values show the Pearson correlation between predicted and observed expressions given by each method for each data set. Largest values are in boldface. GEARS and SAMS-VAE are not applicable to Sciplex2 due to non-binary perturbations. CPA and SAMS-VAE are not applicable to the multi-gene perturbation data due to incompatible internal data preprocessing steps.

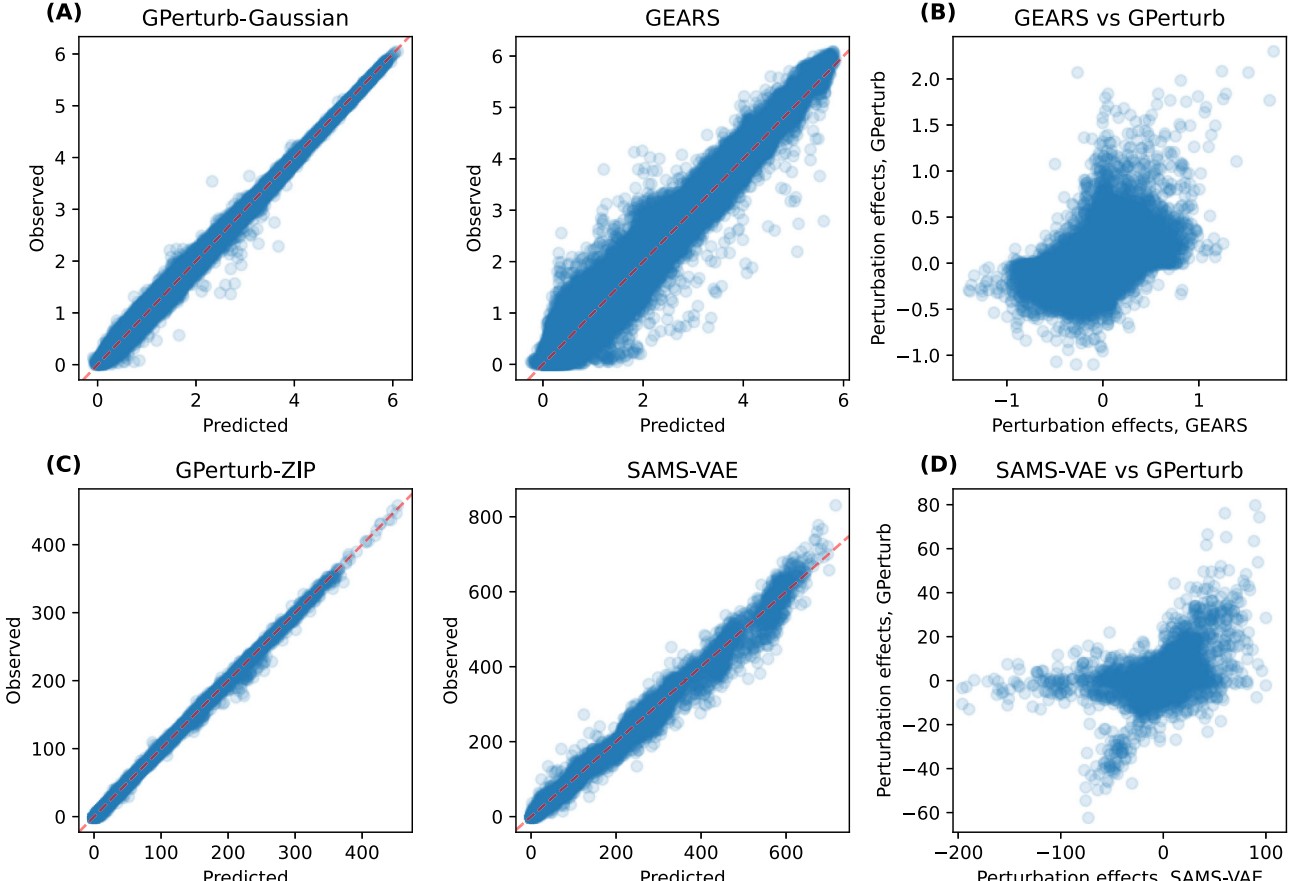

**Fig. 2 | Single-gene perturbation analysis. A** Comparison of predicted versus observed expression from **GPerturb** and GEARS using continuous expression inputs and **B** comparison of predicted perturbation effects. **C** Comparison of predictions from **GPerturb** and SAMS-VAE on count-based expression inputs and **D** comparison of predicted perturbation effects. Source data of the figure are provided as a Source Data file.

## Dosage-based perturbations

We next considered the SciPlex2 dataset[2], where we examined a subset of A549 cells treated with one of the four compounds: dexamethasone (Dex), Nutlin-3a (Nutlin), BMS-345541 (BMS), or vorinostat (SAHA)

across seven different doses. As a benchmark we conducted an analysis using CPA[13], which requires four inputs for each prediction: the cell property, a perturbation type, the expression profile of the cell corresponding to that perturbation and the perturbation type for which we want to predict the expression profile. We recorded the averaged counterfactual predictions of the negative control samples (no perturbation) under each of the 28 unique perturbations (4 compounds × 7 dosages) as counterfactual treatments. For **GPerturb** we recorded the averaged predictions (i.e., prediction values averaged over all cells associated with a common perturbation) for each of the 28 unique perturbations. We then compare the averaged predictions associated with all unique perturbations to the averaged observations in Table 1 and found **GPerturb** outperformed two variants of CPA: CPA-mlp and CPA-logsig. The latter enforces monotonicity of the dose-response relationship in its latent space. In the case of **GPerturb**, the superior performance was achieved in the absence of requiring a basal expression profile as input as needed in CPA. Note that comparison with GEARS and SAMS-VAE was not possible since neither account for non-binary perturbations. We then further investigated the ability of **GPerturb** to model the dosage relationships. Figure 6 illustrates that the predicted dosage-dependent expression levels given by **GPerturb** are more aligned to the measured expression values than both CPA variants particularly for non-monotonic dependencies between drug doses and expression levels. In particular, for PDE4D, CDKN1A and

**Table 2 | Proportion of gene-perturbation pairs with agreement on directionality between methods**

| Dataset | Method | CPA | GEARS | GPerturb | SAMS-VAE |
|---|---|---|---|---|---|
| Replogle | GEARS | 0.532 | – | – | – |
| | GPerturb | 0.520 | 0.625 | – | – |
| | SAMS-VAE | 0.519 | 0.658 | 0.679 | – |
| | GPerturb-ZIP | 0.514 | 0.603 | 0.786 | 0.677 |
| Norman | GEARS | 0.578 | – | – | – |
| | GPerturb | 0.535 | 0.494 | – | – |
| | SAMS-VAE | 0.577 | 0.490 | 0.541 | – |
| | GPerturb-ZIP | 0.556 | 0.593 | 0.508 | 0.502 |
| Yao | GEARS | x | – | – | – |
| | GPerturb | x | 0.502 | – | – |
| | SAMS-VAE | x | x | x | – |
| | GPerturb-ZIP | x | 0.498 | 0.589 | x |

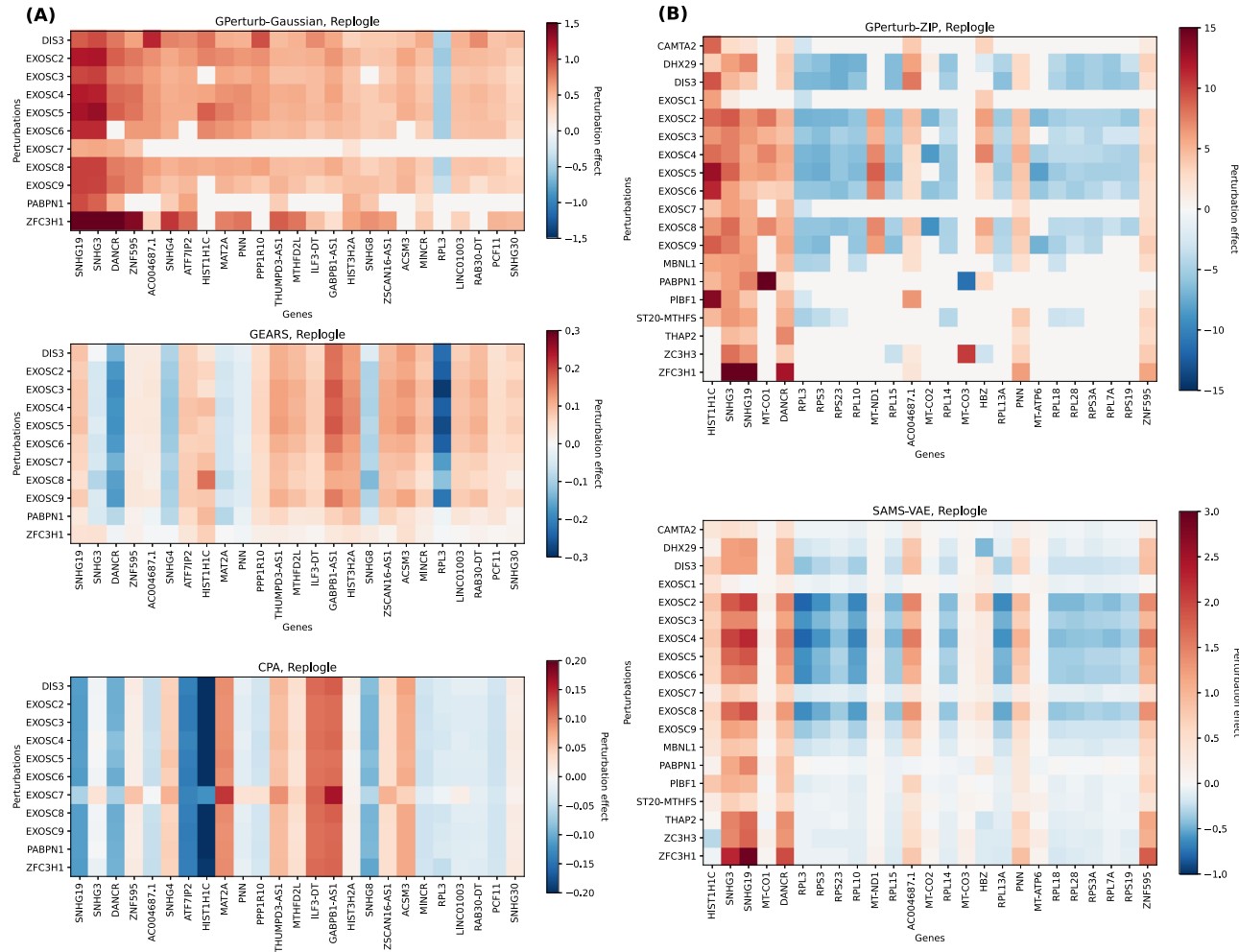

**Fig. 3 | Differences in exosome-related perturbation effects associated with model and pre-processing selection.** Top 25 most differentially expressed genes identified by (**A**) Gaussian and (**B**) Zero-Inflated Poisson versions of **GPerturb** and comparisons to perturbation effects inferred by GEARS, CPA and SAMS-VAE. Note that

for the continuous case, we only include perturbations that are present in the GO graph of the current implementation of GEARS for sake of comparison. The difference in the scales of perturbation effects are due to the different internal data-preprocessing and normalising steps. Source data of the figure are provided as a Source Data file.

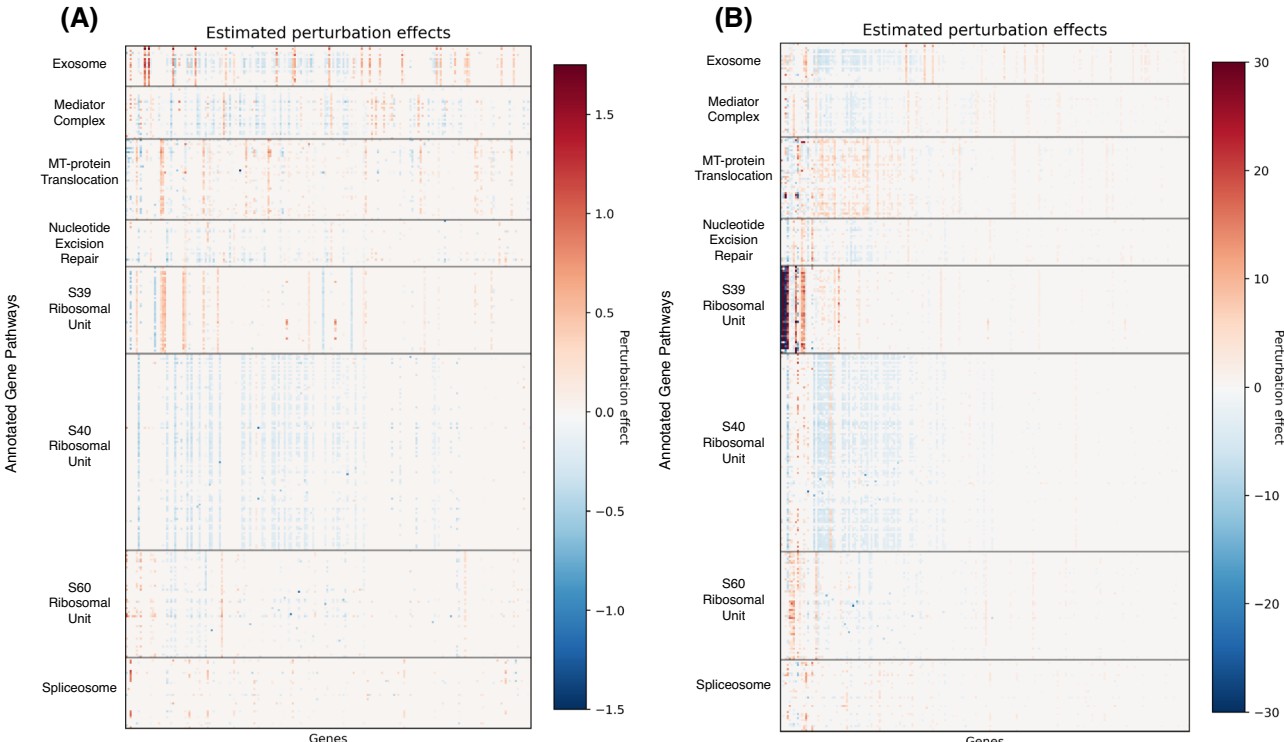

**Fig. 4 | Perturbations by pathway annotations.** Visualisation of 345 perturbations with pathway annotations and grouped by pathway highlights differences between (**A**) Gaussian- and (**B**) ZIP-based **GPerturb**. Each row corresponds to the perturbation effects of a unique perturbation. The perturbation effects for a gene is included only if the associated posterior inclusion probability greater than 0.95. Source data of the figure are provided as a Source Data file.

MDM2, expression varies non-monotonically for BMS, which is not captured by the monotonicity constrained CPA-logsig.

A feature of the semiparametric model specification of **GPerturb** is that it can be used to identify distinct dosage response patterns by examining the gradient information of the estimated perturbation effects. We computed the integrated squared derivatives of the perturbation effects with respect to the dosage level exactly and efficiently using automatic differentiation within **GPerturb**. Large values of this metric allows us to identify genes that are the most sensitive to the dosage of perturbation while low values show no response at all. Examples are illustrated in Fig. 7A. Note that this derivative-based metric captures both monotonic and non-monotonic dosage dependent behaviours. Figure 7B shows the distribution of the derivative metric for each drug on the log scale. In Fig. 7B, only a fraction of genes show high sensitivity to each drug, making it a useful metric for discovery. Figure 7C illustrates example genes which exhibited sensitivity to multiple drugs.

**Comparisons to linear models**

In this section, we apply **GPerturb** to a LUHMES neural progenitor cell CROP-seq dataset[28], and compare its performance to GSFA. This study targets 14 neurodevelopmental genes, including 13 autism risk genes, in LUHMES human neural progenitor cells. The resulting dataset consisting of $N = 8708$ samples and $P = 6000$ selected genes. The perturbations were encoded as one-hot vectors of length 14, each element corresponding to one of the 14 targeted neurodevelopmental genes (i.e., 14 distinct perturbations). The cell information is a real vector of length 4 (`lib_size`: number of total UMI counts, `n_features`: number of genes with non-zero UMI readings, `mt_percent`: percentage of mitochondrial gene expression and `batch`: batch ID). In addition to the one-hot perturbations, the dataset also consists of negative control gRNAs whose perturbations are encoded as zeros. For our proposed method, we randomly select 20% of the dataset as the test set, and use the rest to train **GPerturb**. For GSFA, the results are obtained based on the recommended settings[28].

We further note that in the original study[28], the authors first removed cell level information from the continuous expression inputs by regressing the expression data on the cell information and then apply GSFA to the corresponding standardized residual matrix. In contrast, **GPerturb** disentangles and estimates cell-level and perturbation-induced variations simultaneously, and does not require any additional standardisation. We provided input into **GPerturb** with and without this standardisation but note that the analysis is more interpretable on the original form as additional transformations can affect prediction power and quality.

Figure 8A illustrates the predictions with **GPerturb** on the original data scale and after standardisation. While **GPerturb** shows good predictive performance without the GSFA standardisation applied to the data, it achieves similar correlative performance to GSFA when the data standardisation is used (Pearson correlation $r_{\text{GPerturb}} = 0.248$, $r_{\text{GSFA}} = 0.182$).

We also applied **GPerturb**-Gaussian to a primary human CD8+ T cells dataset[28] in a similar fashion. This study targets 20 genes associated with the T cell response, in both stimulated and unstimulated T cells. The processed dataset consists of $N = 24{,}955$ samples and $P = 6000$ genes. The perturbations were encoded as one-hot vectors of length 20, which correspond to the 20 targeted genes in the study, and cell information was provided as a real vector of length 5 (`lib_size`: number of total UMI counts, `n_features`: number of genes with non-zero UMI readings, `mt_percent`: percentage of mitochondrial gene expression, `donor`: T Cell donor ID and `stimulated`: whether or not the T Cell is stimulated).

In GSFA a modification is used to capture differences in perturbation effects between stimulated and unstimulated cells. We replicated the modification in **GPerturb** model to accommodate potentially different perturbation effects for stimulated and

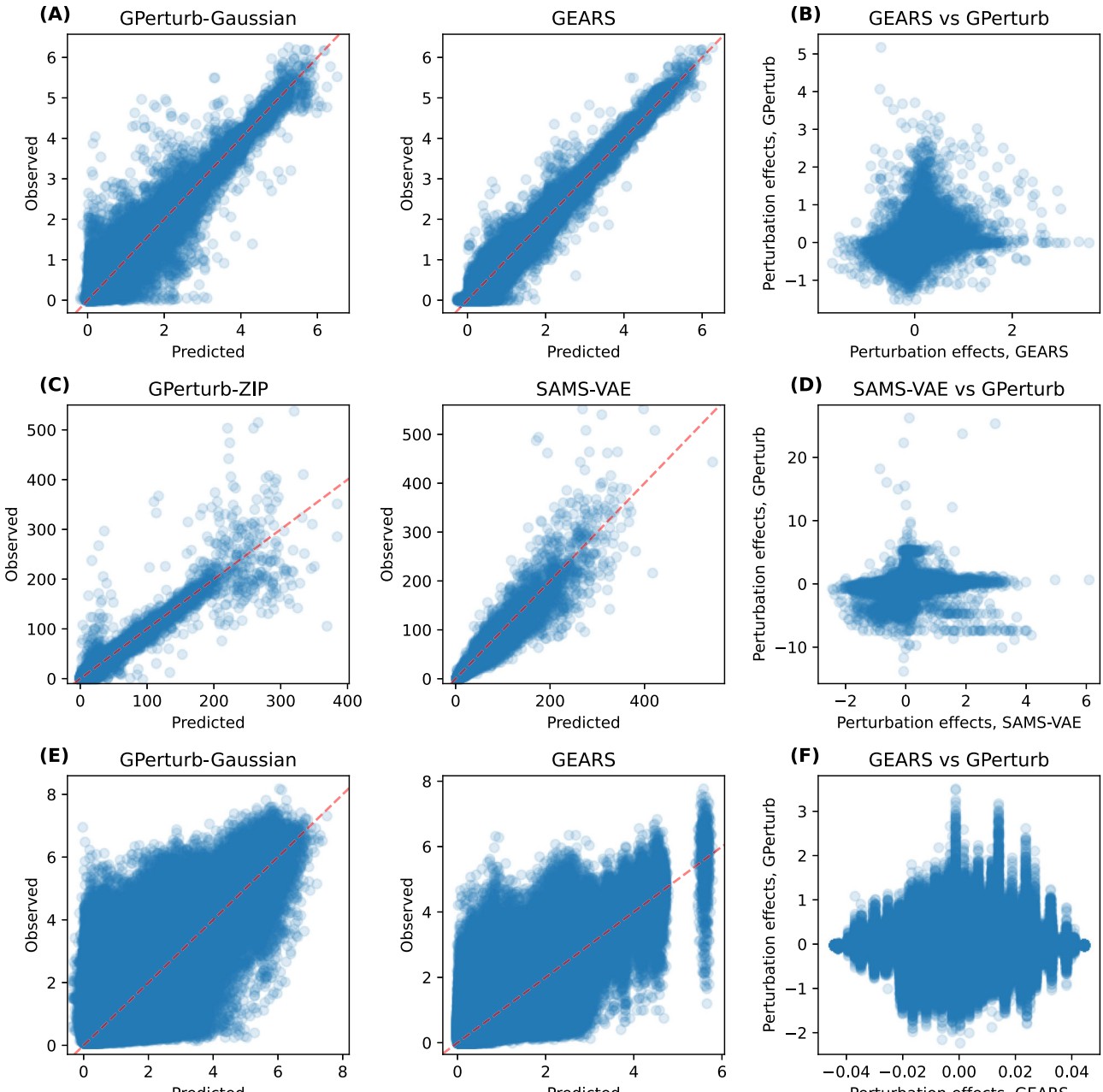

**Fig. 5 | Multi-gene perturbation analysis. A** Comparison of predicted and observed expression from **GPerturb** and GEARS using continuous expression inputs and **B** corresponding comparison of perturbation effects. **C** Comparison of predicted and observed expression from **GPerturb**-ZIP and SAMS-VAE using count-based expression inputs and **D** corresponding comparison of perturbation effects. **E** Comparison of predicted and observed expression from **GPerturb** and GEARS using continuous expression input and **F** corresponding comparison of perturbation effects. Source data of the figure are provided as a Source Data file.

unsimulated T Cells. Similar to the previous example, we randomly select 20% of the dataset as the test set, and use the rest to train GPerturb. We report the fitted results in Fig. 8D. When comparing the predictive performance, GSFA showed greater correlation than **GPerturb** on the standardised data ($r_{GPerturb} = 0.271$, $r_{GSFA} = 0.335$). However, Fig. 8C shows that the standardisation applied by GSFA was likely to be disadvantageous and unnecessary with **GPerturb** since it can be applied without the initial cell information regression step.

## Discussion

There are a number of state-of-the-art single-cell perturbation modelling methods currently available (including many not directly considered here), but a detailed analysis of the pre-processing, training and inference requirements of each method highlights significant differences in the approach and requirements associated with each method. While there has been considerable interest in deep learning based approaches, **GPerturb** adopts a more classical non-linear regression based modelling strategy, which provides a non-deep learning approach to support model training and prediction by focusing on directly modelling individual genes rather than via the use of latent representations in many other recent methods. Our analysis shows that **GPerturb** is capable of attaining state of the art performance despite these significant design differences and is highly versatile and computationally efficient (see Supplementary Table 1). A feature showed in our experimental results (Table 1) is that **GPerturb** in both forms could be applied to all four examples, while other methods could only be used for a subset of these. This highlights the versatility of **GPerturb** as it is able to handle single, multi-gene,

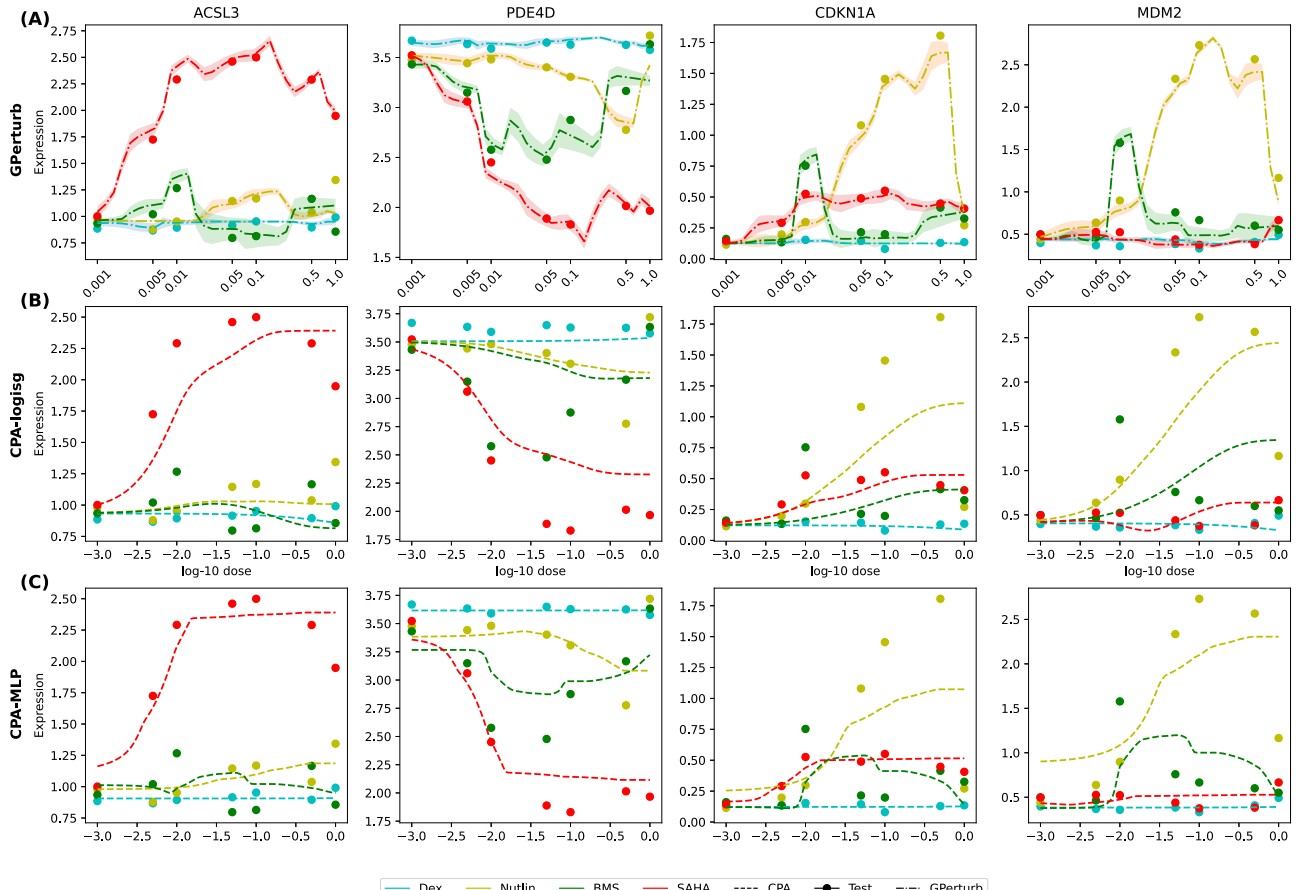

**Fig. 6 | Analysis of continuous dosage-based perturbations.** Predicted dosage-linked expression levels given by (**A**) **GPerturb**, (**B**) monotonically-constrained CPA and (**C**) unrestricted CPA on selected genes from the Sciplex2 dataset[2]. Different colours are assigned to the four drugs (Dex, Nutlin, BMS, SAHA). Dotted lines correspond to the estimated means given by different methods. Shaded regions are the corresponding 95% credible band given by GPerturb. Source data of the figure are provided as a Source Data file.

categorical and continuous perturbation inputs. While predictive performance derived from retrospective analysis of existing data sets is an extremely important metric, it is important to note that validation on independent experiments is vital.

Our experiments show that direct performance comparisons between methods must be interpreted carefully and may not always be applicable. For example, comparisons between GSFA and other methods were not shown since GSFA operates and returns results in terms of standardised input data residuals. In addition, we also tried to compare the estimated sparse perturbation effects given by **GPerturb** with SAMS-VAE and CPA, but found no straightforward way to do so due to the fact that **GPerturb** directly estimates sparse perturbation effects on the gene level, while CPA and SAMS-VAE focus on finding sparse low-dimensional embeddings of them. Furthermore, since our proposed **GPerturb** framework allows handling of both continuous normalised and count-based data using Gaussian and zero-inflated Poisson based likelihoods, we have observed that while the alternate versions of **GPerturb** attain comparable prediction accuracy with methods using comparable input data, the perturbation effects captured by the Gaussian and ZIP versions of **GPerturb** could be different. This highlights that variations in data processing and modelling could affect the conclusions drawn from the same raw data and adds a further source of uncertainty on the true validity of any biological insights drawn from perturbation prediction methods.

Our experiments are consistent with other recent more extensive evaluation studies[25,26,35], which have also found that prediction performance is highly context-dependent and that no single method excels across all scenarios. These evaluations include recently developed single-cell foundation models, which can also be applied for perturbation effect prediction. In some cases, performance of these foundation models may be no better than simple linear models[26]. The scalable Gaussian Process regression models we have introduced in **GPerturb** provide a highly-effective and complementary approach for single-cell perturbation modelling. These models can be of utility for direct prediction tasks or as a methodologically distinct benchmark for the development of new methods. Future work could examine extensions of this Gaussian Process framework as a credible non-deep learning based approach for handling multi-omics or spatially resolved molecular data.

## Methods

### GPerturb-Gaussian

We first discuss the model with **X** being a matrix of pre-processed continuous responses. We will give a counting data version of the model alongside with a schematic illustration of the additive modelling structure later.

Let $k_{\nu_\mu}, k_{\nu_\gamma}, k_{\nu_\eta} : \mathbb{R}^L \times \mathbb{R}^L \to \mathbb{R}$ be Gaussian process kernels governed by kernel parameters $\nu_\mu, \nu_\gamma, \nu_\eta$, respectively. Let $g_\mu, g_\gamma, g_\eta : \mathbb{R}^L \to \mathbb{R}$ be the mean functions of the corresponding Gaussian processes.

We first define the gene-level additive perturbation model as follows:

$$m_p : \mathbb{R}^D \to \mathbb{R}; \quad \lambda_p \in \mathbb{R}; \quad (1)$$

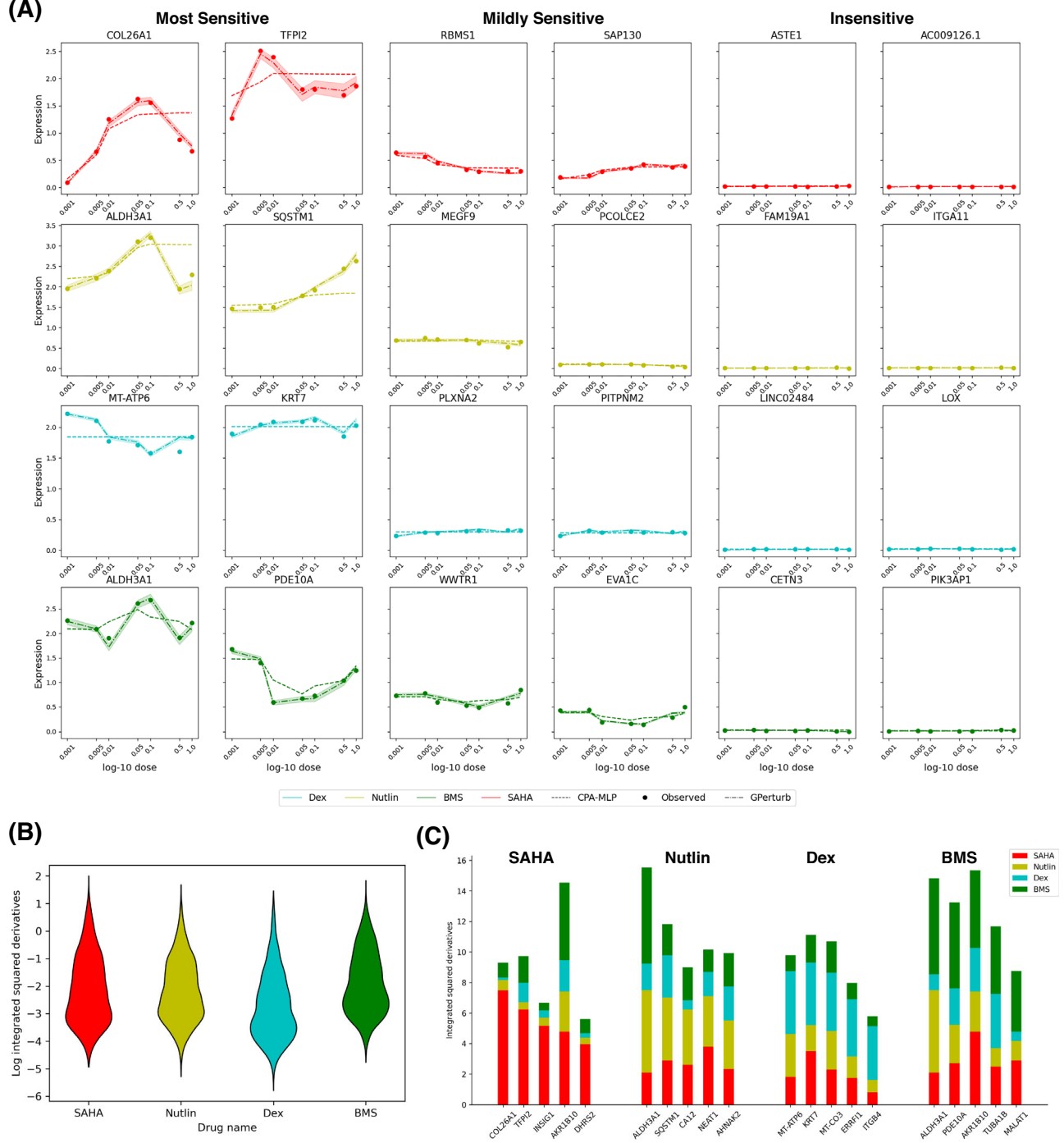

**Fig. 7 | Sensitivity to perturbations. A** Dosage vs Estimated perturbed expression. Each row corresponds to one of the four drugs (Vorinostat, Nutlin-3a, dexamethasone, BMS-345541). Left two columns consist of most sensitive genes measured in integral of squared derivative, mid two columns correspond to genes showing medium sensitivity (derivative metric around 1), and the right two columns show genes with low sensitivity (derivative metric around 0). Dots corresponds to the observed expressions under different gene-perturbation pairs averaged over the test set. Dotted lines correspond to the estimated means. Shaded regions are the corresponding 95% credible band given by GPerturb. **B** Violin plot of the log of integrals of squared derivatives for all 5000 genes in the Sciplex2 dataset. Different colours correspond to different drugs. **C** Bar plots of the sum of derivative metrics of the top 5 genes most sensitive to each of the four drugs. Note that AKR1B10 and ALDH3A1 are sensitive to more than one drugs. Source data of the figure are provided as a Source Data file.

$$\mu_p \sim \mathcal{GP}(g_\mu, k_{\nu_\mu}); \quad \gamma_p \sim \mathcal{GP}(g_\gamma, k_{\nu_\gamma}); \tag{2}$$

$$\eta_p \sim \mathcal{GP}(g_\eta, k_{\nu_\eta}); \quad z_{ip} \sim \text{Bernoulli}\left(\sigma(\eta_p(\mathbf{C}_i))\right); \tag{3}$$

$$X_{ip} \sim \mathcal{N}\left(m_p(\mathbf{K}_i) + z_{ip}\mu_p(\mathbf{C}_i), \log(\exp(\lambda_p + z_{ip}\gamma_p(\mathbf{C}_i)) + 1)\right), \tag{4}$$

where $\sigma(x) = \frac{1}{1+\exp(-x)}$. In this model setup, $m_p$ is a fixed but unknown function that takes the cell-level information vector $\mathbf{K}_i$ associated with the $i$th sample as input, and returns $m_p(\mathbf{K}_i)$ as the expected basal

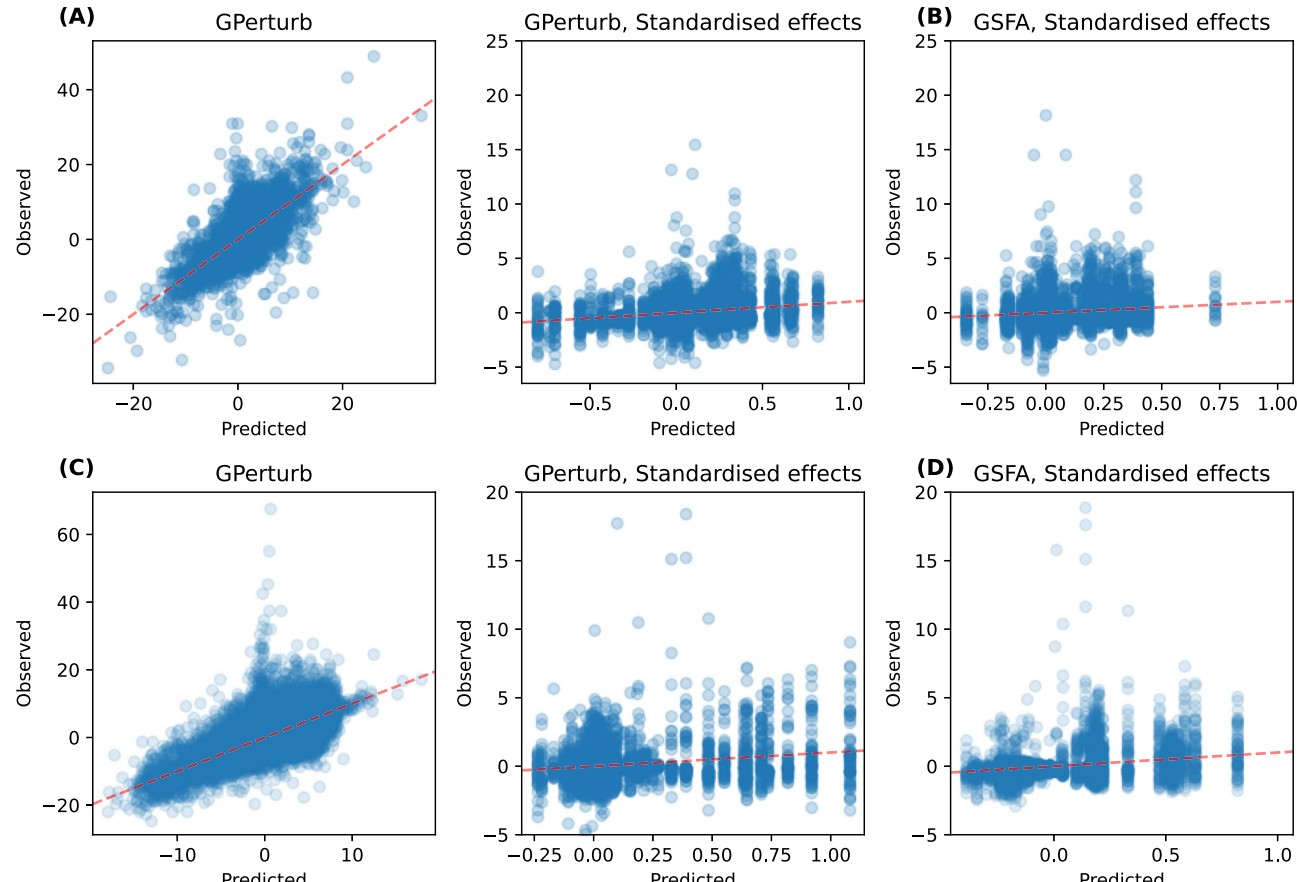

**Fig. 8 | Comparison to GSFA. A** Comparison of predicted and observed expression for LUHMES neural progenitor cell CROP-seq using **GPerturb** with and without GSFA-based standardisation and **B** GSFA reported predicted and observed standardised expression. **C** Comparison of predicted and observed expression for primary human CD8+ T cells datasets[28] given by **GPerturb** with and without GSFA-based standardisation and **D** GSFA reported predicted and observed standardised expression. Source data of the figure are provided as a Source Data file.

expression level of the $p$th gene in the $i$th sample. $\lambda_p$ is the basal variability parameter of the expression level of the $p$th gene shared across all samples $i = 1, ..., N$. $z_{ip}$ is a binary toggle controlling whether or not the expression level of the $p$th gene in the $i$th sample is perturbed by the perturbation vector $\mathbf{C}_i$. The success probability of $z_{ip}$ depends on $\mathbf{C}_i$ through $\eta_p(\mathbf{C}_i)$, a random function $\eta_p$ evaluated at $\mathbf{C}_i$ (Note that under our additive modelling setup, the binary toggles $z_{ip}$ are the same for all cells receiving the same perturbation). $\mu_p, \gamma_p$ are also random functions that take $\mathbf{C}_i$ associated with the $i$th sample as input, and return $\mu_p(\mathbf{C}_i), \gamma_p(\mathbf{C}_i)$ as the potential mean- and variability-level perturbation effects on the expression level of the $p$th gene in the $i$th sample. Schematic illustration and graphical representation of the Gaussian model is given in Supplementary Fig. 1.

We then assume the observed perturbed expression level $X_{ip}$ associated with perturbation $\mathbf{C}_i$ and cell-level information $\mathbf{K}_i$ follows a Gaussian distribution with mean being the sum of basal mean $m_p(\mathbf{K}_i)$ and mean-level perturbation effect $z_{ip}\mu_p(\mathbf{C}_i)$, and variance being a positive function of the sum between the common basal variability $\lambda_p$ and a variability-level perturbation $z_{ip}\gamma_p(\mathbf{C}_i)$. We choose to use the function $\log(\exp(\cdot)+1)$ to map the unconstrained variability parameters to the positive real variance parameters since it is approximately linear when the magnitude of input is large, and therefore would partially retain the additive structure between the basal states and perturbation effects in a similar fashion to the mean parameters in comparison with e.g., $\exp(\cdot)$.

Under our modelling setup, the parameters naturally partition into two groups: the random perturbation-specific parameters $\{\mu_p, \gamma_p, \eta_p, \{z_{ip}\}_{i=1}^N\}_{p=1}^P$, and the unknown but fixed basal level

parameters $\{m_p, \lambda_p\}_{p=1}^P$. We do not treat the basal level parameters as random variables since the primary objective of the model is to learn how the gene expression levels $X_{ip}$ respond to different perturbations. In the proposed model, the basal states only play the role of "intercept" or "offset", and is not of primary interest to us. In addition, treating the basal level parameters as unknown but fixed model parameters also simplifies the inference procedure and reduces the computational cost of the proposed model.

**GPerturb-ZIP**

We now discuss the model for expression count data. Similar to the continuous model, we define the count data based gene-level additive perturbation model using a zero-inflated Poisson likelihood as follows

$$m_p : \mathbb{R}^D \to \mathbb{R}; \quad \mu_p \sim \mathcal{GP}(g_\mu, k_{\nu_\mu}); \quad \eta_p \sim \mathcal{GP}(g_\gamma, k_{\nu_\gamma}); \quad (5)$$

$$\pi_p \in (0,1); \quad z_{ip} \sim \text{Bernoulli}\left(\sigma(\eta_p(\mathbf{C}_i))\right); \quad (6)$$

$$X_{ip} \sim \text{ZIP}\left(\log(\exp(m_p(\mathbf{K}_i) + z_{ip}\mu_p(\mathbf{C}_i)) + 1), \pi_p\right), \quad (7)$$

where $\mu_p, m_p, \eta_p, z_{ip}$ have the same interpretation as in the deviance-based model, $\pi_p$ is the proportion of excessive zeros on the $p$th gene shared across all samples $i = 1, ..., N$, and $\text{ZIP}(\mu, \pi)$ denotes a zero-inflated Poisson distribution with expected Poisson rate $\mu$ and probability of excessive zeros $\pi$. Note that our ZIP model does not aim to estimate the pattern of excessive zeros of the dataset. Hence

the quantity $z_{ip}\mu_p(\mathbf{C}_i))$ should be interpreted as "the conditional perturbation effect given that the corresponding observation $X_{ip}$ is not an excessive zero". Schematic illustration and graphical representation of the Zero-inflated Poisson model is given in Supplementary Fig. 2.

We also considered handling potential over-dispersion by modelling $\mathbf{X}$ using a zero-inflated Gamma-Poisson likelihood (a different parameterisation of Negative Binomial). However, we find the Gamma-Poisson model and Poisson model achieved similar level of prediction performance on real datasets, and the majority of estimated dispersion parameters are far less than 1, showing no strong sign of over-dispersion. Hence we focus on the Poisson model in this section for sake of simplicity. The details of the zero-inflated Gamma-Poisson model can be found in Supplementary Information.

For both the deviance-based Gaussian and the Zero-inflated Poisson model, we recommend setting $k_\mu$, $k_\gamma$, $k_\eta$ to be RBF kernels $k(x_1, x_2) = \nu^{(1)} \exp(-\nu^{(2)}||x_1 - x_2||_2^2)$ governed by kernel parameters $\nu_\mu^{(1)}, \nu_\gamma^{(1)}, \nu_\eta^{(1)} = 1$, $\nu_\mu^{(2)}, \nu_\gamma^{(2)}, \nu_\eta^{(2)} = 0.1$, $g_\mu = g_\gamma = 0$ and $g_\eta = -3$ as these prior specifications give satisfactory results in all of our numerical experiments. These choices of priors reflect our belief that all $\mu_p(\mathbf{C}_i)$ and $\gamma_p(\mathbf{C}_i)$ have the same marginal prior $\mathcal{N}(0, 1)$, and the prior on $\sigma(\eta_p(\mathbf{C}_i))$, the inclusion probability of perturbation effect of $\mathbf{C}_i$ on the $p$th gene, is concentrated at around 0.05. Alternative choices are also discussed in the following sections.

## Posterior inference

In this section, we discuss the posterior inference strategy of the proposed models. We first give the posterior inference procedure of the deviance-based model. Let $\lambda = \{\lambda_p\}_{p=1}^P$. Let $p\left(\{\mu_p(\mathbf{C}_i), \gamma_p(\mathbf{C}_i), \eta_p(\mathbf{C}_i), z_{ip}\}_{i,p=1}^{N,P}\right)$ be the prior of the associated perturbation-specific parameters. Let

$$p\left(\mathbf{X}|\mathbf{C}, \mathbf{K}, \{\mu_p(\mathbf{C}_i), \gamma_p(\mathbf{C}_i), \eta_p(\mathbf{C}_i), z_{ip}, m_p(\mathbf{K}_i)\}_{i,p=1}^{N,P}, \lambda\right)$$
$$= \prod_{i,p=1}^{N,P} \mathcal{N}\left(X_{ip}; m_p(\mathbf{K}_i) + z_{ip}\mu_p(\mathbf{C}_i), \log(\exp(\lambda_p + z_{ip}\gamma_p(\mathbf{C}_i)) + 1)\right) \quad (8)$$

be the likelihood of the observed gene-expression level matrix $\mathbf{X}$ given the perturbation matrix $\mathbf{C}$, the cell-level information matrix $\mathbf{K}$, and all model parameters. Since the number of samples $N$ and the number of genes $P$ are usually large, jointly estimating the posterior $p\left(\{\mu_p(\mathbf{C}_i), \gamma_p(\mathbf{C}_i), \eta_p(\mathbf{C}_i), z_{ip}\}_{i,p=1}^{N,P}|\mathbf{X}, \mathbf{C}, \mathbf{K}, \{m_p(K_i)\}_{i,p=1}^{N,P}, \lambda\right)$ is computationally infeasible. We therefore use amortised variational inference[36] to address this issue: Let $f_\xi : \mathbb{R}^L \to \mathbb{R}^{6P}$ be a neural network parameterized by a real vector $\xi$. We approximate the full posterior $p\left(\{\mu_p(\mathbf{C}_i), \gamma_p(\mathbf{C}_i), \eta_p(\mathbf{C}_i), z_{ip}\}_{i,p=1}^{N,P}|\mathbf{X}, \mathbf{C}, \mathbf{K}, \{m_p(K_i)\}_{i,p=1}^{N,P}, \lambda\right)$ using the following variational posterior:

$$q_\xi\left(\{\mu_p(\mathbf{C}_i), \gamma_p(\mathbf{C}_i), \eta_p(\mathbf{C}_i), z_{ip}\}_{i,p=1}^{N,P}\right) = \prod_{i,p=1}^{N,P}\left(\mathcal{N}\left(\mu_p(\mathbf{C}_i); f_\xi^{(p)}(\mathbf{C}_i), \exp(f_\xi^{(p+P)}(\mathbf{C}_i))\right)\right.$$
$$\times \mathcal{N}\left(\gamma_p(\mathbf{C}_i); f_\xi^{(p+2P)}(\mathbf{C}_i), \exp(f_\xi^{(p+3P)}(\mathbf{C}_i))\right)$$
$$\times \mathcal{N}\left(\eta_p(\mathbf{C}_i); f_\xi^{(p+4P)}(\mathbf{C}_i), \exp(f_\xi^{(p+5P)}(\mathbf{C}_i))\right)$$
$$\left.\times \text{Bernoulli}\left(z_{ip}; \sigma(\eta_p(\mathbf{C}_i))\right)\right), \quad (9)$$

where $f_\xi^{(p)}(\mathbf{C}_i)$ denotes the $p$th entry of $f_\xi(\mathbf{C}_i)$, $\mathcal{N}(\cdot; \mu, s^2)$ denotes a Gaussian p.d.f. with mean $\mu$ and variance $s^2$, and $\text{Bernoulli}(\cdot; \pi)$ denotes a Bernoulli p.m.f. with success probability $\pi$. Similarly, for the fixed but unknown basal level functions $\{m_p\}_{p=1}^P$, we let $f_\phi : \mathbb{R}^D \to \mathbb{R}^P$ be a neural network parameterized by a real vector $\phi$, and use $f_\phi^{(p)}(\mathbf{K}_i)$ to parameterize $m_p(\mathbf{K}_i)$ for all $i = 1, ..., N$ and $p = 1, ..., P$. The evidence

lower bound (ELBO) of the deviance-based model then takes the form

$$\text{ELBO}_G(\xi, \phi, \lambda; \mathbf{X}, \mathbf{C}, \mathbf{K}) = E_{q_\xi}\left(\log p(\mathbf{X}|\mathbf{C}, \mathbf{K}, \{\mu_p(\mathbf{C}_i), \gamma_p(\mathbf{C}_i), \eta_p(\mathbf{C}_i), z_{ip}, f_\phi^{(p)}(K_i)\}_{i,p=1}^{N,P}, \lambda)\right)$$
$$- KL\left(q_\xi\left(\{\mu_p(\mathbf{C}_i), \gamma_p(\mathbf{C}_i), \eta_p(\mathbf{C}_i), z_{ip}\}_{i,p=1}^{N,P}\right),\right.$$
$$\left. p\left(\{\mu_p(\mathbf{C}_i), \gamma_p(\mathbf{C}_i), \eta_p(\mathbf{C}_i), z_{ip}\}_{i,p=1}^{N,P}\right)\right), \quad (10)$$

where $E_{q_\xi}$ denotes expectation with respect to the variational posterior $q_\xi$. We estimate the variational posterior $q_\xi$ and all other model parameters by maximizing (an empirical version of) $\text{ELBO}_G(\xi, \phi, \lambda; \mathbf{X}, \mathbf{C}, \mathbf{K})$. The Bernoulli random variables in Eq. (9) is approximated using Gumbel softmax[37].

Let $\{\xi^*, \phi^*, \lambda^*\} = \arg\max_{\xi,\phi,\lambda} \text{ELBO}_G(\xi, \phi, \lambda; \mathbf{X}, \mathbf{C}, \mathbf{K})$. Once the model has been fitted, we can then construct both approximate point and interval estimates of parameters of our interest. For example, let $\mathbf{C}'$ be a generic perturbation vector. One can form approximate point or interval estimates of the posterior inclusion probability $\sigma(\eta_p(\mathbf{C}'))$, which controls if the expression level of the $p$th gene is perturbed by $\mathbf{C}'$, using the variational posterior $q_{\xi^*}(\eta_p(\mathbf{C}')) = \mathcal{N}\left(f_{\xi^*}^{(p+4P)}(\mathbf{C}'), \exp(f_{\xi^*}^{(p+5P)}(\mathbf{C}'))\right)$. Compared with LFSR[38] used in GSFR[28], identifying perturbation effects using posterior inclusion probability is more intuitive and interpretable thanks to the full Bayesian framework of the proposed model.

We now discuss the posterior inference of the Zero-inflated Poisson model. It can be parameterised and estimated in a similar fashion to the deviance-based model: Let $\pi = \{\pi_p\}_{p=1}^P$. Let $p\left(\{\mu_p(\mathbf{C}_i), \eta_p(\mathbf{C}_i), z_{ip}\}_{i,p=1}^{N,P}\right)$ be the prior of the associated perturbation-specific parameters in the Zero-inflated Poisson model. Let

$$p\left(\mathbf{X}|\mathbf{C}, \mathbf{K}, \{\mu_p(\mathbf{C}_i), \eta_p(\mathbf{C}_i), z_{ip}, m_p(\mathbf{K}_i)\}_{i,p=1}^{N,P}, \pi\right)$$
$$= \prod_{i,p=1}^{N,P} \text{ZIP}\left(X_{ip}; \log(\exp(m_p(\mathbf{K}_i) + z_{ip}\mu_p(\mathbf{C}_i)) + 1), \pi_p\right) \quad (11)$$

be the likelihood of the raw counting data $\mathbf{X}$, where $\text{ZIP}(\cdot; \mu, \pi)$ is the p.m.f. of a Zero-Inflated Poisson distribution with Poisson rate $\mu$ and probability of excessive zeros $\pi$. Similar to the deviance-based model, let $f_\theta : \mathbb{R}^L \to \mathbb{R}^{4P}$ be a neural network parameterised by a real vector $\theta$. Let

$$q_\theta\left(\{\mu_p(\mathbf{C}_i), \eta_p(\mathbf{C}_i), z_{ip}\}_{i,p=1}^{N,P}\right) = \prod_{i,p=1}^{N,P}\left(\mathcal{N}\left(\mu_p(\mathbf{C}_i); f_\theta^{(p)}(\mathbf{C}_i), \exp(f_\theta^{(p+P)}(\mathbf{C}_i))\right)\right.$$
$$\times \mathcal{N}\left(\eta_p(\mathbf{C}_i); f_\theta^{(p+2P)}(\mathbf{C}_i), \exp(f_\theta^{(p+3P)}(\mathbf{C}_i))\right)$$
$$\left.\times \text{Bernoulli}\left(z_{ip}; \sigma(\eta_p(\mathbf{C}_i))\right)\right), \quad (12)$$

be the variational posterior of the perturbation-specific parameters $\{\mu_p(\mathbf{C}_i), \eta_p(\mathbf{C}_i), z_{ip}\}_{i,p=1}^{N,P}$. As in the deviance-based model, we use $f_\phi(K_i)$ to parameterize the basal level parameter $m_p(K_i)$ for all $i = 1, ..., N$ and $p = 1, ..., P$. Then the ELBO of the Zero-inflated Poisson model $\text{ELBO}_P(\theta, \phi, \pi; \mathbf{X}, \mathbf{C}, \mathbf{K})$ is defined in a similar fashion to Eq. (10), and the model parameters are estimated by maximizing (an empirical version of) $\text{ELBO}_P(\theta, \phi, \pi; \mathbf{X}, \mathbf{C}, \mathbf{K})$ with respect to $\{\theta, \phi, \pi\}$.

## Magnitudes of perturbation vectors

In our proposed methods, the perturbation vector $\mathbf{C}_i$ can either be binary (indicating the presence of a perturbation) or continuous (representing e.g., dosage). When $\mathbf{C}_i$ represents the continuous dosage of a perturbation, we expect that the potential perturbation effects is positively correlated to the dosage (at least in a sensible range before some ceiling effects). Similarly, when $\mathbf{C}_i = \mathbf{0}$ (i.e., no perturbation at all), we expect there is no potential perturbation effects. To impose these physical constraints, we recommend modifying the model and

inference procedure as follows (Here we focus on the deviance based model. The zero-inflated Poisson model can be modified in a similar fashion): We first replace the standard RBF kernels on $v_\mu$, $v_\gamma$ by a modified "zero-passing" RBF kernel[39]. This modification ensures that samples $\{\mu_p, \gamma_p\}_{p=1}^P$ drawn from the modified Gaussian process prior would satisfy $\{\mu_p(\mathbf{0}) = \gamma_p(\mathbf{0}) = 0\}_{p=1}^P$. We also replace the generative process of $z_{ip}$ with $z_{ip} \sim \mathrm{Bernoulli}\left(\mathbb{1}(\|\mathbf{C}_i\|_2^2 \neq 0)\sigma(\eta_p(\mathbf{C}_i))\right)$, which ensures that no $z_{ip}$ would be triggered when the input $\mathbf{C}_i = \mathbf{0}$. This choice also reflects our prior belief that even though the potential effects of a perturbation $\mathbf{C}_i$ depends on its magnitude, whether or not a gene is perturbed only depends on the presence of the perturbation (i.e., $\|\mathbf{C}_i\|_2^2 \neq 0$), but not the scale of it. Similar constraints have also been used previously[13,14].

In addition to the generative process, we also adjust the inference procedure accordingly. We modify the variational posterior in Eqn. (9) as follows:

$$q'_\xi\left(\{\mu_p(\mathbf{C}_i), \gamma_p(\mathbf{C}_i), \eta_p(\mathbf{C}_i), z_{ip}\}_{i,p=1}^{N,P}\right) = \prod_{i,p=1}^{N,P} \left(\mathcal{N}\left(\mu_p(\mathbf{C}_i); \|\mathbf{C}_i\|_2 f_\xi^{(p)}(\mathbf{C}_i), \|\mathbf{C}_i\|_2^2 \exp(f_\xi^{(p+P)}(\mathbf{C}_i))\right) \right.$$
$$\times \mathcal{N}\left(\gamma_p(\mathbf{C}_i); \|\mathbf{C}_i\|_2 f_\xi^{(p+2P)}(\mathbf{C}_i), \|\mathbf{C}_i\|_2^2 \exp(f_\xi^{(p+3P)}(\mathbf{C}_i))\right) \times \mathcal{N}\left(\eta_p(\mathbf{C}_i); f_\xi^{(p+4P)}(\mathbf{C}_i), \exp(f_\xi^{(p+5P)}(\mathbf{C}_i))\right)$$
$$\left. \times \mathrm{Bernoulli}\left(z_{ip}; \mathbb{1}(\|\mathbf{C}_i\|_2^2 \neq 0)\sigma(\eta_p(\mathbf{C}_i))\right)\right).$$
$$(13)$$

Here we rescale the variational posterior of the mean- and viability-level perturbation by a factor $\|\mathbf{C}_i\|_2$. This ensures that both terms would be zero when there is no perturbation, and would explicitly depend on the size of $\mathbf{C}_i$ otherwise. We also modify the variational distribution of $z_{ip}$ in the same way as in the generative process. These modification ensures that both generative process and posterior inference are inline with the physical constraints discussed above. We use this modified generative process and inference procedure as our default model setup for the rest of the paper.

### Single gene perturbation

The Perturb-Seq dataset[33] was pre-processed and filtered using the previously described pre-processing steps[14,40]. The resulting dataset $\mathbf{X} \in \mathbb{N}^{N \times P}$ consists of counting data of $N = 118,461$ cells and $P = 1187$ genes. For $i = 1, ..., N$, the perturbation $\mathbf{C}_i \in \mathbb{R}^L$ is either a length $L = 722$ one-hot vector, representing one of the 722 unique CRISPR guides (perturbations), or a zero vector, representing the perturbation associated with negative controls (non-targeting CRISPR guides). The non-targeting negative controls are treated as the baseline level. The cell information $\mathbf{K}_i \in \mathbb{R}^D$ is a length $D = 4$ real vector (lib_size: total number of UMI counts, n_features: number of genes with non-zero UMI readings, mt_percent: percentage of mitochondrial gene expression, scale_factor: core scale factor).

### Multigene perturbation prediction

We compared GPerturb's performance on predicting multi-gene perturbation outcomes with the knowledge-graph informed GEARS using the Perturb-seq dataset[15,34]. We followed a previously described data-preprocessing process[15]. The resulting dataset $\mathbf{X} \in \mathbb{R}^{N \times P}$ consists of $N = 89,357$ cells and $P = 5045$ genes. For $i = 1, ..., N$, the perturbation $\mathbf{C}_i \in \mathbb{R}^L$ is a length $L = 103$ binary vector where the positions of ones encode the perturbed genes. (The dataset consists of 131 two-gene perturbations.) The cell information $\mathbf{K}_i \in \mathbb{R}^D$ is a length $D = 2$ real vector (lib_size: total number of UMI counts, n_features: number of genes with non-zero UMI readings). We randomly select 20% of the dataset as the test set, and use the rest as training set. For both our GPerturb and GEARS, the recommended settings are used to fit the models.

We further compared GPerturb with GEARS using the multiplexed Perturb-seq dataset[12] using the same procedure described above. We

follow the data-preprocessing process given by the authors. The resulting dataset $\mathbf{X} \in \mathbb{R}^{N \times P}$ consists of $N = 24,192$ cells and $P = 15,668$ genes. For $i = 1, ..., N$, the perturbation $\mathbf{C}_i \in \mathbb{R}^L$ is a length $L = 600$ binary vector where the positions of ones encode the perturbed genes. The cell information $\mathbf{K}_i \in \mathbb{R}^D$ is a length $D = 3$ real vector (lib_size: total number of UMI counts, n_features: number of genes with non-zero UMI readings and mt_percent: percentage of mitochondrial gene expression). We compare the performance of GEARS and GPerturb under the same setting described above.

### SciPlex2

The SciPlex2 dataset[2] (GSM4150377) consists of A549 cells treated with one of the four compounds: dexamethasone (Dex), Nutlin-3a (Nutlin), BMS-345541 (BMS), or vorinostat (SAHA) across seven different doses. We follow previously described data pre-processing steps[13]. The resulting dataset $\mathbf{X} \in \mathbb{R}^{N \times P}$ consists of $N = 20,643$ cells and $P = 5000$ genes. For $i = 1, ..., N$, the perturbation $\mathbf{C}_i \in \mathbb{R}^L$ is a length $L = 4$ vector with only one non-zero entry whose position and value encode the compound type and dosage, respectively. Similar to the previous sections, the perturbation associated with negative controls are encoded as $\mathbf{C}_i = \mathbf{0}$ and treated as the baseline level. The cell information $\mathbf{K}_i \in \mathbb{R}^D$ is a length $D = 2$ real vector (lib_size: total number of UMI counts, n_features: number of genes with non-zero UMI readings). We randomly select 20% of the dataset as the test set, and use the rest to train GPerturb. For both Gaussian GPerturb and CPA, the recommended settings are used to fit the models.

### Derivative metrics for identification of dosage patterns

We denote $\hat{D}_{i,j}(x)$ as the estimated perturbation effect of perturbation $j$ on gene $i$ at dosage level $x$. Due to the semiparametric specification of GPerturb, we can compute $\frac{d}{dx}\hat{D}_{i,j}(x)$, the derivative of the perturbation effects with respect to the dosage level $x$, exactly and efficiently (thanks to automatic differentiation) and use the derivative information to capture interesting perturbation patterns.

We identify genes that are the most sensitive to the dosage of perturbation $j$ by investigating the integral of the squared derivative $\hat{D}_i^j = \int_{A_{\min}^j}^{A_{\max}^j} \left(\frac{d}{dx}\hat{D}_{i,j}(x)\right)^2 dx$ for each gene $i$ in the data set, where $A_{\min}^j, A_{\max}^j$ are the minimum and maximum dosage of perturbation $j$ respectively.

We choose the integral of the squared derivative as a measure of sensitivity since this quantity equals zero if and only if $\hat{D}_{i,j}(x)$ equals some constant, indicating that the perturbation effect does not depend on dosage at all, and is large only if the magnitude of rate of change in the perturbation effect is large over the interval $[A_{\min}^j, A_{\max}^j]$.

### Reporting summary

Further information on research design is available in the Nature Portfolio Reporting Summary linked to this article.

## Data availability

The Sciplex2 dataset with associated metadata were obtained from https://www.ncbi.nlm.nih.gov/geo/query/acc.cgi?acc=GSM4150377. The raw Perturb-Seq dataset from Replogle et al.[33] was obtained from https://gwps.wi.mit.edu. The raw Perturb-Seq dataset from Norman et al.[34] was obtained from https://dataverse.harvard.edu/api/access/datafile/6154020. The raw Perturb-Seq dataset from Yao et al.[12] was obtained from https://www.ncbi.nlm.nih.gov/geo/query/acc.cgi?acc=GSE221321. All processed data, count matrices and results (including pre-trained models and visualisation) are available at https://figshare.com/articles/dataset/GPerturb_Gaussian_process_modelling_of_single-cell_perturbation_data/26491588. Attribution to elements in Fig. 1A: Red blood cells icon, Gene icon. Source data are provided with this paper.

## Code availability

The code used to develop the model, perform the analyses and generate results in this study is publicly available and has been deposited in Github repository **GPerturb** at https://github.com/hwxing3259/GPerturb, under MIT license. The specific version of the code associated with this publication is archived in Zenodo and is accessible via https://doi.org/10.5281/zenodo.15305114.

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

## Acknowledgements

The authors are supported by an EPSRC Turing AI Acceleration Fellowship (Grant Ref: EP/V023233/1).

## Author contributions

H.X. proposed and implemented the method, performed numerical experiments, wrote the initial draft and revised the manuscript. C.Y. supervised the project, revised the initial draft and wrote the final manuscript.

## Competing interests

The authors declare no competing interests.
