## [Transparent Peer Review file · Nature Communications]

GPerturb: Gaussian process modelling of single-cell perturbation data

Corresponding Author: Professor Christopher Yau

Version 0:

Reviewer comments:

Reviewer #1

(Remarks to the Author)

In this work, the authors developed GPerturb, a Gaussian Process-based approach, for modeling and predicting single-cell gene expression after genetic or chemical perturbations. The manuscript is poorly written: missing figure panels, wrong section references, insufficient descriptions of experimental setups and limitations, etc. More importantly, the results are not convincing that GPerturb is outperforming other existing methods in any significant aspects. I thus recommend rejection of this work.

The clarity of this manuscript needs significant improvements:

- Introduction needs a lot more background information for the general audience, especially given readers of Nature Communications are from diverse backgrounds.

- Figure 1:

Panel (a) cartoon is confusing. What is input and output? What do perturbation vectors and cell information vectors represent biologically? Readers need to dig into the method section to understand the current cartoon, which defeats the purpose of having a summary cartoon.

Panel (b) caption reads "Example of observed and predicted gene (FTL) expression values", yet it is unclear which data shown in this panel are observed and which are predicted. Also, what exactly does FTL stand for? Having acronyms that do not correspond to actual words can be quite confusing for readers.

Missing Panel (c), where is the UMAP in the caption?

For panel (d), it'd be helpful to also have boxplots and statistics shown. Also, according to caption this is only showing subset of genes predicted from given perturbation. How did the authors select the subset of genes to plot? And are these perturbations chosen because they were showing the best predicted effects?

- Many supplementary sections are mis-referenced in the manuscript.

The results comparing performances of different methods in Figure 2 and Table 1 are not convincing that GPerturb has significant improvements over other existing methods. It's interesting to see GPerturb captures more non-monotonic relationships than another method, Compositional Perturbation Autoencoder (CPA), in Figure 2c. However, are those real biological signals GPerturb is better at modeling, or is GPerturb better at overfitting to experimental noises? How about cases when GPerturb perform worse than CPA?

It's also interesting seeing different data preprocessing led to drastically different outputs from GPerturb (Figure 2d), but then which one is more biologically relevant? Are there any recommendations for how users should preprocess data before using GPerturb?

One of the important reasons for doing multiplexed perturbation is to capture nonlinear genetic interactions from perturbations, is GPerturb able to model that?

Lastly, it would be very interesting to see what happens when applying GPerturb to highly multiplexed perturb-seq dataset from Yao et al. (PMID: 37872410). Is GPerturb still able to model and predict perturbation outcomes when most single cells receive >10 perturbations?

(Remarks on code availability)

Given the current state of this manuscript, reviewing code is not necessary at this point.

Reviewer #2

(Remarks to the Author)

The authors present a novel approach for predicting perturbations from multiplex single cell data. The article is very well written and the methods are explained in detail, along with comprehensive further experiments and testing on synthetic data in the supplementary materials. The approach has good performance and provides improvements over existing methods in terms of interpretability. There are some questions regarding the overall approach that could be addressed in the manuscript:

- Given the use of neural networks in the amortised variational inference approach, have the authors explored the variation in predictions between multiple training runs of the same model, and could this partially explain the difference between the continuous and ZIP predictions?
- It is surprising that the Gamma Poisson distribution does not improve on the Poisson, could the authors give a more precise idea of the magnitude of the dispersion parameters (or perhaps the ratio of variance to the mean)?
- The authors mention that the predictions made by the continuous version of the model differ from those made by the ZIP version, despite achieving similar accuracy. Is there a preferred approach between the two (continuous or UMI count input)? Would the authors ever recommend using both?

Minor comments:

- Could the authors provide a reference for amortised variational inference?
- Typos: Page 1, line 32, "to flexibility", page 6 line 140 "Eentries", page 7 line 194 "on on", page 8 line 215 "armortized"
- Several missing section references, page 7 line 198, page 8 line 205, others.

(Remarks on code availability)

Reviewer #3

(Remarks to the Author)

In this paper by Xing and Yau, the authors introduce GPerturb, a gaussian process and Bayesian framework to model and predict perturbations from multiplex single-cell perturbation.

Overall this is one more tool in the field of single-cell perturbation prediction and analysis. In contrast to other novel tools, GPerturb does not use an embedding, deep learning approach. Instead it uses Gaussian processes to model expression functions.

Major issues:

it was unclear to me how GPerturb outperforms other tools in the field. It seems to me, from the figures that GPerturb is at least as good as the tools that the authors use to compare. So, what do we gain with GPerturb?

Also, it was unclear to me how the Bayesian aspect of GPerturb enhances the results. The authors do not explain in much detail the Bayesian component of the tool.

How does GPerturb compare with other tools in terms of computational costs?

Minor issues:

There are several errors throughout:

E.g. "studies" is repeated twice in the first sentences.

Many acronyms are undefined, such as GSFA, SAMS-VAE, GEARS, etc. Given the broad readership of this journal these acronyms should be explained some.

(Remarks on code availability)

Code was readable, well-written. I could install the code and run the examples.

Version 1:

Reviewer comments:

Reviewer #1

(Remarks to the Author)

The clarity of the manuscript has improved significantly. I respectfully ask authors to please sanity check manuscript before submitting to avoid simple mistakes such as in line 120-121 that references main figure panels which do not exist, among many other minor ones that I will not bother listing here.

It's exciting to see that GPerturb is applicable to a variety of Perturb-seq data. I appreciate the authors' efforts in addressing my previous comments to demonstrate the applicability of GPerturb to different scenarios. However, the ask for "biological relevance" is still not addressed. One advantage of GPerturb that the authors advertised for is its interpretability, and such biological insights are going to be very important for readers of Nature Communications. What additional biological insights can researchers uncover with GPerturb? Predictive performance cannot be the only evaluation metric as the model could simply be predicting well by overfitting to the noises inherent to each dataset / experiment. And the authors do not have any evaluation on the "interpretability" part of the model. How do we know if the model is interpreting true biological signal rather than interpreting random noises in the data? Especially the pre-processing step seem to greatly affect model's interpretability, does that mean the interpretation is inherently associated to noises from how the data is pre-processed?

(Remarks on code availability)

Really glad seeing authors providing notebooks to reproduce figures.

Reviewer #2

(Remarks to the Author)

The authors have addressed my concerns, and improved the presentation of the paper. I agree with the authors' assertion that the value of the method is in demonstrating it is possible to produce valuable results using statistical approaches, when the majority of state of the art methods are using deep learning models.

(Remarks on code availability)

The code is clearly documented, however there are some minor problems installing and running the code - the numpy version needs to be fixed to 1.x, and there is a typo in the code provided in the README, in one example the function is written as GPerturb_gaussian, when it should be GPerturb_Gaussian

The provision of code for reproducibility as well as all of the data used is good, but there appear to be more minor issues that would mean the code needs modifying to run with the package provided, or at least there should be more detailed instruction (e.g. what is GPerturb_model_1?)

Reviewer #3

(Remarks to the Author)

Overall, the authors have addressed my concerns in a thorough manner and should be commended for this. The method however provides incremental advances over other tools thus diminishing my enthusiasm for the work. In my opinion this work would be of interest to a more specialized methods audience.

(Remarks on code availability)

The code is easy to follow and read, wish examples provided for users.

Version 2:

Reviewer comments:

Reviewer #1

(Remarks to the Author)

I highly appreciate the authors' efforts in addressing all my previous comments. The quality of this manuscript has notably improved. However, I still cannot recommend this work for publication in Nature Communications. The computational method presented in this work offers only modest innovation to the field. The overall impact could be strengthened by having novel biological insights with thorough orthogonal validation, which is also missing in this work.

(Remarks on code availability)

The authors have provided detailed instructions for installing and running the code, along with step by step notebook for reproducing figures.

Reviewer #2

(Remarks to the Author)

(Remarks on code availability)

The updated code and instructions are now clearer and straightforward to follow to reproduce the work in the manuscript.

We thank the reviewers for their constructive and insightful comments. The following details our response to reviewers' questions and comments.

The additional analyses and discussions detailed below are also included in the updated supplementary materials.

Reviewer #1

In this work, the authors developed GPerturb, a Gaussian Process-based approach, for modelling and predicting single-cell gene expression after genetic or chemical perturbations. The manuscript is poorly written: missing figure panels, wrong section references, insufficient descriptions of experimental setups and limitations, etc. More importantly, the results are not convincing that GPerturb is outperforming other existing methods in any significant aspects.

We thank the reviewer for the constructive comments, and we address the concerns in our response below.

Concerns regarding clarity and presentation of the manuscript

We have rewritten the manuscript, considerably extending the length of the paper and provided a more complete exposition of the methodology and experiments.

The results comparing performances of different methods in Figure 2 and Table 1 are not convincing that GPerturb has significant improvements over other existing methods.

We agree with Reviewer #1 that GPerturb does not always outperform existing state-of-the-art methods and different methods appear to be optimal for different contexts. However, in all four experiments, all versions of GPerturb performed well. These findings are consistent with a number of recent evaluation studies which we reference in our new Discussion section.

We would like to clarify that attaining state-of-the-art prediction accuracy is not the only objective of our proposed GPerturb. GPerturb demonstrates that state of the art prediction results can be achieved without the use of deep learning approaches and the use of a regression based approach can deliver more interpretable results without sacrificing prediction power in comparison with existing methods. With similar level of computation cost (See also Table 1) and prediction performance, our more interpretable and conceptually simpler GPerturb can be more advantageous for downstream work in comparison with existing method, thanks to the gene-level (instead of latent space level) additive architecture and the sparsity constraints GPerturb puts directly on the perturbation effects (instead of latent embeddings).

It's interesting to see GPerturb captures more non-monotonic relationships than another method, Compositional Perturbation Autoencoder (CPA), in Figure 2c. However, are those real biological signals GPerturb is better at modeling, or is GPerturb better at overfitting to experimental noises? How about cases when GPerturb perform worse than CPA?

The results we reported in original submission Fig 2c was chosen to reproduced the dosage response analysis given by Lotfollahi et al⁵ (Supplementary Fig S5) and their online tutorial. In

EXPRESSION INPUT TYPE	APPROACH	DATASET			
		Sciplex2 ¹	Single-gene perturba- tion ²	Multi-gene perturba- tion ³	Multi-gene perturba- tion ⁴
Continuous, transformed	GPerturb-Gaussian	1.65×10^3	1.35×10^4	7.82×10^4	3.78×10^3
	CPA-logsig	1.38×10^3	2.27×10^4	1.01×10^5	-
	CPA-MLP	1.40×10^3	2.28×10^4	1.03×10^5	-
	GEARS	-	1.17×10^4	5.94×10^4	4.29×10^3
Count-based	GPerturb-ZIP	1.60×10^3	1.32×10^4	8.80×10^4	3.71×10^3
	GPerturb-ZIGP	1.64×10^3	1.38×10^4	8.87×10^4	3.82×10^3
	SAMS-VAE	-	2.09×10^4	6.55×10^4	-

Table 1: Comparison of wall clock running time. Values show the averaged wall clock running time in seconds for each method and each data set over 3 repetitions.

the original submission we compared our GPerturb with the default CPA whose dosage response curves are constrained to be monotonic log-sigmoid functions (CPA-logsig). We added comparison with a variant of CPA¹ whose dosage response curves are parameterised as more flexible neural networks (CPA-MLP) without the monotonicity constraint. We report the predicted dosage-linked expression levels on selected genes given by different models and the corresponding observed expressions averaged over the test set in Fig 1. Visually we see that the estimates given by CPA-MLP are in better agreement with both the observations and the GPerturb estimates than CPA-logsig. We then investigate the prediction performance of CPA and GPerturb quantitatively by comparing the Pearson correlation between the observed and predicted perturbed expressions averaged over the test set for each of the the unique gene-perturbation pairs. We found that GPerturb attains the best performance ($r_{\text{GPerturb}} = 0.988$), and the more flexible CPA-MLP achieves better performance ($r_{\text{CPA-MLP}} = 0.985$) than CPA-logsig ($r_{\text{CPA-logsig}} = 0.980$). The results agree with our visual demonstration in Fig 1. We further applied the more flexible CPA-MLP to other examples, and see that the additional model flexibility does improve model fit (Table 2) in most of the cases. Since both our visual and quantitative comparisons are based on test sets that were not seen by the models during the training phase, we conclude that introducing more flexible modeling architecture (e.g. relaxing the monotonic constraint) does improve model fit and allows both CPA and GPerturb to capture more non-monotonic relationships.

It’s also interesting seeing different data pre-processing led to drastically different outputs from GPerturb (Figure 2d), but then which one is more biologically relevant? Are there any recommendations for how users should preprocess data before using GPerturb?

To demonstrate the difference between the discrete and continuous GPerturb, we report the estimated perturbation effects given by different models studied in the paper using the Replogle et al² dataset in Fig 2. Since CPA focuses on counterfactual prediction for individual cells and does not provide standalone perturbation effect estimates, for each perturbation, we choose to use the averaged difference between the counterfactual predictions and the corresponding ob-

¹This can be activated by selecting the `mlp` option for the `doser.type` switch.

Figure 1: SciPlex2 dataset. Predicted dosage-linked expression levels.. Predicted dosage-linked expression levels given by GPerturb, CPA-logsig and CPA-MLP on selected genes from the Sciplex2¹ dataset. Different colours are assigned to the four drugs (Dex, Nutlin, BMS, SAHA). Dots corresponds to the observed expressions under different gene-perturbation pairs averaged over the test set. Shaded regions are the corresponding 95% credible band given by GPerturb. Genes are selected to reproduce Supplementary Fig S5 in Lotfollahi et al⁵.

EXPRESSION INPUT TYPE	APPROACH	DATASET			
		Sciplex2 ¹	Single-gene perturba- tion ²	Multi-gene perturba- tion ³	Multi-gene perturba- tion ⁴
Continuous, transformed	GPerturb-Gaussian	0.988	0.981	0.979	0.798
	CPA-logsig	0.980	0.986	0.975	-
	CPA-MLP	0.985	0.984	0.977	-
	GEARS	-	0.977	0.982	0.802
Count-based	GPerturb-ZIP	0.973	0.972	0.961	0.861
	GPerturb-ZIGP	0.920	0.970	0.960	0.831
	SAMS-VAE	-	0.944	0.952	-

Table 2: Comparison of predictive performance. Values show the Pearson correlation between predicted and observed expressions on test set given by each method for each data set. GEARS and SAMS-VAE are not applicable to Sciplex2 due to non-binary perturbations. CPA and SAMS-VAE are not applicable to the multi-gene perturbation data from Yao et al⁴ due to incompatible internal data pre-processing steps.

served “basal” expressions over all control-group cells in the test set as a proxy of the estimated perturbation effect given by CPA. From Figure 2 we see the estimations given by ZIP-GPerturb agrees with SAMS-VAE⁶, but the results given Gaussian-GPerturb, GEARS and CPA differ from each other, even though all these methods achieve good prediction performance (Table 2).

The disagreement between the continuous models shown in Fig 2 could be originated from the different model architectures and data pre-processing steps: The log-transformed continuous data may not necessarily follow the Gaussian assumption used in GEARS, CPA and GPerturb-Gaussian (to see this, note that log-transformed count data are non-negative, a feature can not be captured by the Gaussian modeling assumption without further constraint). As a result, the misspecified likelihood can make the training and inference of these methods more sensitive to the choice modeling architectures and data-preprocessing steps, and potentially lead to different results in Fig 2. In contrast, the discrete models use more appropriate likelihood (i.e. modeling count data using zero-inflated Poisson or Negative Binomial) for UMI count data, and we see that these models return consistent results (Fig 2, right).

In addition, the difference between perturbation effects given by the continuous and discrete models in our example may be associated with the different interpretations of perturbation effects: the continuous Gaussian model measures perturbation effects in term of fold changes on log scale relative to the basal states, while the count-based model measures perturbation effects in term of additive changes in UMI counts. Both modeling approach are sensible and have been widely used in practice. However, we would like to highlight that existing methods such as CPA, GEARS and SAMS-VAE focus on a specific type of data (either discrete or continuous), and develop highly specialised model and training procedure for the chosen data type. As a result, these models are not able to identify and quantify the impact of data pre-processing steps on the estimation results under a common modeling framework. In contrast, our GPerturb can naturally handle different type of data using the appropriate likelihood functions. This allows us

(a) Continuous data

(b) Discrete data

Figure 2: Estimated perturbation effects associated with exosome-related perturbations in Replage et al² on subsets of differentially expressed genes identified by Gaussian and ZIP GPerturb respectively. Note that for the continuous case, we only include perturbations that are present in the GO graph of the current implementation of GEARS for sake of comparison. The difference in the scales of perturbation effects is due to the different internal data-preprocessing and normalising steps.

to reveal the impact of different data pre-processing steps under a common framework. Hence we think of the difference between the discrete and continuous GPerturb as a feature rather than a bug, as it offers users the freedom to choose and compare different data types and pre-processing steps.

In practice, we recommend users to experiment with different data pre-processing steps and see to what degree do the resulting estimations depend on them. However, the log-transformed continuous data may not necessarily follow the Gaussian model assumption. From this perspective, the Poisson or Gamma-Poisson models based on the raw counting data are preferable as they may introduce less model misspecification and give more robust results.

Figure 3: Scatter plot of averaged perturbed expression predictions vs observations for each multiplexed perturbation in Yao et al⁴.

One of the important reasons for doing multiplexed perturbation is to capture nonlinear genetic interactions from perturbations, is GPerturb able to model that?
 Lastly, it would be very interesting to see what happens when applying GPerturb to highly multiplexed perturb-seq dataset from Yao et al. (PMID: 37872410). Is GPerturb still able to model and predict perturbation outcomes when most single cells receive more than 10 perturbations?

To demonstrate GPerturb’s capability of learning nonlinear genetic interactions from multiplexed perturbations, we further compare GPerturb and GEARS using the highly multiplexed Yao et al (PMID: 37872410)⁴ dataset. CPA and SAMS-VAE are not applicable to the dataset due to incompatible internal data pre-processing steps. We report scatter plots of estimated vs observed expression levels on test set given by Gaussian GPerturb and GEARS in Fig 3. The two methods achieve similar level of accuracy (See also Table 2, confirming GPerturb’s capability of predicting multiplexed perturbation effects.

Reviewer #2

The authors present a novel approach for predicting perturbations from multiplex single cell data. The article is very well written and the methods are explained in detail, along with comprehensive further experiments and testing on synthetic data in the supplementary materials. The approach has good performance and provides improvements over existing methods in terms of interpretability. There are some questions regarding the overall approach that could be addressed in the manuscript

We thank the reviewer for the comments. We address the concerns in our response below.

Figure 4: Estimated perturbation effects of exosome-related perturbations in Replegle et al² on the top-25 most differentially expressed genes identified by GPerturb. **Top row: Results given by Gaussian GPerturb. Bottom row: Results given by ZIP GPerturb.**

Given the use of neural networks in the amortised variational inference approach, have the authors explored the variation in predictions between multiple training runs of the same model, and could this partially explain the difference between the continuous and ZIP predictions?

To examine the variability of GPerturb over multiple runs, we repeatedly run both Gaussian and ZIP-GPerturb independently five times under different random seeds, and find that both both Gaussian and ZIP-GPerturb give robust results under repeated runs. We demonstrate this in Fig 4 by comparing the estimated effects of exosome-related perturbations in Replegle et al² dataset given by the first run (Fig 4 left) with the averaged estimation based on 5 independent runs (Fig 4 right) under different random seeds. We see the results from a single run roughly agree with the averaged results, confirming robustness of GPerturb under repeated runs.

It is surprising that the Gamma Poisson distribution does not improve on the Poisson, could the authors give a more precise idea of the magnitude of the dispersion parameters (or perhaps the ratio of variance to the mean)?

We compare and report the prediction performance of ZIP and ZIGP GPerturb in Table 2. We

see the ZIP model consistently outperforms the ZIGP model in term of prediction accuracy in Pearson correlation. In addition, we also report the estimated log overdispersion parameters for each of the dataset in Fig 5. We found that over 45%, 98%, 79%, and 95% of the estimated gene-level log overdispersion parameters are less than 0 for the Sciplex2¹, Replogle et al², Norman et al³ and Yao et al⁴ datasets respectively. To further inspect the model fit of the Sciplex2 dataset who has the highest proportion of positive log overdispersion parameters, we report in Fig 6 the scatter plot of the estimated verses observed counts on Sciplex2 for both ZIP- and ZIGP-GPerturb. Again we see ZIP-GPerturb better predicts observed counts in comparison with ZIGP-GPerturb (See also Table 2). Hence we conclude that the more parsimonious ZIP-GPerturb is preferable over ZIGP-GPerturb for the numerical examples we considered in this paper.

The authors mention that the predictions made by the continuous version of the model differ from those made by the ZIP version, despite achieving similar accuracy. Is there a preferred approach between the two (continuous or UMI count input)? Would the authors ever recommend using both?

To demonstrate the difference between the discrete and continuous GPerturb, we also report the estimated perturbation effects given by different models studied in the paper using the Replogle et al² dataset in Fig 2. Since CPA focuses on counterfactual prediction for individual cells and does not provide standalone perturbation effect estimates, for each perturbation, we choose to use the averaged difference between the counterfactual predictions and the corresponding observed “basal” expressions over all control-group cells in the test set as a proxy of the estimated perturbation effect given by CPA. From Figure 2 we see the estimations given by ZIP-GPerturb agrees with SAMS-VAE⁶, but the results given Gaussian-GPerturb, GEARS and CPA differ from each other, even though all these methods achieve good prediction performance (Table 2).

The disagreement between the continuous models shown in Fig 2 could be originated from the different model architectures and data pre-processing steps: The log-transformed continuous data may not necessarily follow the Gaussian assumption used in GEARS, CPA and GPerturb-Gaussian (to see this, note that log-transformed count data are non-negative, a feature can not be captured by the Gaussian modeling assumption without further constraint). As a result, the misspecified likelihood can make the training and inference of these methods more sensitive to the choice modeling architectures and data-preprocessing steps, and potentially lead to different results in Fig 2. In contrast, the discrete models use more appropriate likelihood (i.e. modeling count data using zero-inflated Poisson or Gamma-Poisson) for UMI count data, and we see that these models return consistent results (Fig 2, right).

In addition, the difference between the perturbation effects given by the continuous and discrete models in our example may be associated with the different interpretations of perturbation effects: the continuous Gaussian model measures perturbation effects in term of fold changes on log scale relative to the basal states, while the count-based model measures perturbation effects in term of additive changes in UMI counts. Both modeling approach are sensible and have been widely used in practice. However, we would like to highlight that existing methods such as CPA, GEARS and SAMS-VAE focus on a specific type of data (either discrete or continuous), and develop highly specialised model and training procedure for the chosen data type. As a result, these models are not able to identify and quantify the impact of data pre-processing steps on the estimation results under a common modeling framework. In contrast, our GPerturb can naturally handle different type of data using the appropriate likelihood functions. This allows us

Figure 5: Histograms of estimated log dispersion parameters for each of the datasets we studied in the paper.

Figure 6: Non-zero observed counts for each cell-gene pair vs corresponding estimated mean for each cell-gene pair given by ZIP and ZIGP GPerturb on test set of Sciplex2 dataset¹.

to reveal the impact of different data pre-processing steps under a common framework. Hence we think of the difference between the discrete and continuous GPerturb as a feature rather than a bug, as it offers users the freedom to choose and compare different data types and pre-processing steps.

In practice, we recommend users to experiment with different data pre-processing steps and see to what degree do the resulting estimations depend on them. However, the log-transformed continuous data may not necessarily follow the Gaussian model assumption. From this perspective, the Poisson or Gamma-Poisson models based on the raw counting data are preferable as they may introduce less model misspecification and give more robust results.

Minor concerns regarding writing and presentation.

We thank the reviewer for the suggestions. The concerns are addressed in the updated manuscript.

Reviewer #3

In this paper by Xing and Yau, the authors introduce GPerturb, a gaussian process and Bayesian framework to model and predict perturbations from multiplex single-cell perturbation.

Overall this is one more tool in the field of single-cell perturbation prediction and analysis. In contrast to other novel tools, GPerturb does not use an embedding, deep learning approach. Instead it uses Gaussian processes to model expression functions.

We thank the reviewer for the comments. We address the concerns in our response below.

It was unclear to me how GPerturb outperforms other tools in the field. It seems to me, from the figures that GPerturb is at least as good as the tools that the authors use to compare. So, what do we gain with GPerturb?

We agree with Reviewer #3 that GPerturb does not outperform existing state-of-the-art methods by a large margin. However, we would like to clarify that attaining state-of-the-art prediction accuracy is not the only objective of our proposed GPerturb. In particular, we would like to highlight that GPerturb is capable of delivering more interpretable results without sacrificing prediction power in comparison with existing methods: Note that in contrast with the latent embedding strategy used in e.g. GEARS⁷, SAMS-VAE⁶ and CPA⁵, GPerturb adopts a both conceptually and computationally simpler gene-level additive framework with nonlinear components which balances interpretability and flexibility. A recent work from Ahlmann-Eltze et al⁸ demonstrated that such additive architecture is able to achieve competitive predictive performance in comparison with models with larger scales and more complicated architectures such as GEARS and CPA, which is confirmed by our numerical studies (Table 2). This broadly similar performance is not surprising as all methods are built on very powerful deep-learning based modeling machinery, and the high level of intrinsic noise in single-cell screening data prevents any computational methods from perfectly recovering the underlying biological signals. With similar level of computation cost and prediction performance, our more explainable GPerturb can be more advantageous for downstream work in comparison with existing method, thanks to the gene-level (instead of latent space level) additive architecture and the sparsity constraints GPerturb puts directly on the perturbation effects (instead of latent embeddings).

Also, it was unclear to me how the Bayesian aspect of GPerturb enhances the results. The authors do not explain in much detail the Bayesian component of the tool.

The probabilistic Bayesian modeling framework improves interpretability and uncertainty quantification of the results. SAMS-VAE and CPA focus on predicting counterfactual perturbed gene expressions using latent embeddings which lack natural biological meaning. GEARS is able to predict the effect of a given perturbation on individual genes, and give ad-hoc uncertainty quantification. However, the lack of sparsity in GEARS predictions makes it difficult for users to identify genes responsive to different perturbations in a principled way. Our probabilistic

Figure 7: Estimated perturbation effects associated with exosome-related perturbations in Replogle et al² on a subset of differentially expressed genes identified by the model under different posterior inclusion probability thresholds. Note that as the threshold increases, less gene-perturbation pairs are deemed to be responsive.

Bayesian modeling framework address these issues. For example, GPertrub provides easy-to-interpret uncertainty quantification (credible intervals) of the predicted perturbation effects (See Fig 1, Upper row). In addition, GPerturb is able to identify and select responsive genes to different perturbations in a straightforward and interpretable fashion using the estimated posterior inclusion probabilities (PIP) for each gene-perturbation pair (i.e. the probability of a gene being responsive to a perturbation) thanks to the probabilistic modeling framework (See Methods). To further illustrate this point, we report the estimated perturbational effects of exosome-related perturbations on the top-25 most differentially expressed genes identified by GPerturb in Replogle et al² data at different PIP inclusion thresholds (Fig 7). We see that by increasing the PIP inclusion threshold from 0.5 to 0.99, users can easily filter out the subset of gene-perturbation pairs that are most confidently selected by the model based on this interpretable threshold. This means users can use GPerturb to directly handle queries such as “how likely is a gene responsive to a given perturbation” or “what subset of genes are most likely to be responsive to a given perturbation” without any ad-hoc post-processing steps used in e.g. GEARS and CPA.

How does GPerturb compare with other tools in terms of computational costs?

Since distinct modeling architectures are used in different methods, we choose to report the wall clock running time as a fair benchmark for computation cost. Results are reported in Table 1. We see that the running time of discrete and continuous GPerturb are similar, and the computation cost of GPerturb in term of running time is on a similar level to existing methods. All experiments are conducted on our machine with an AMD Ryzen 7 2700 CPU and an NVidia RTX 2060 GPU.

Minor concerns regarding writing and presentation.

We thank the reviewer for the suggestions. The concerns are addressed in the updated manuscript.

References

- [1] Sanjay R Srivatsan, José L McFaline-Figueroa, Vijay Ramani, Lauren Saunders, Junyue Cao, Jonathan Packer, Hannah A Pliner, Dana L Jackson, Riza M Daza, Lena Christiansen, et al. Massively multiplex chemical transcriptomics at single-cell resolution. *Science*, 367(6473):45–51, 2020.
- [2] Joseph M Replogle, Reuben A Saunders, Angela N Pogson, Jeffrey A Hussmann, Alexander Lenail, Alina Guna, Lauren Mascibroda, Eric J Wagner, Karen Adelman, Gila Lithwick-Yanai, et al. Mapping information-rich genotype-phenotype landscapes with genome-scale perturb-seq. *Cell*, 185(14):2559–2575, 2022.
- [3] Thomas M Norman, Max A Horlbeck, Joseph M Replogle, Alex Y Ge, Albert Xu, Marco Jost, Luke A Gilbert, and Jonathan S Weissman. Exploring genetic interaction manifolds constructed from rich single-cell phenotypes. *Science*, 365(6455):786–793, 2019.
- [4] Douglas Yao, Loic Binan, Jon Bezney, Brooke Simonton, Jahanara Freedman, Chris J Frangieh, Kushal Dey, Kathryn Geiger-Schuller, Basak Eraslan, Alexander Gusev, et al. Scalable genetic screening for regulatory circuits using compressed perturb-seq. *Nature Biotechnology*, pages 1–14, 2023.
- [5] Mohammad Lotfollahi, Anna Klimovskaia Susmelj, Carlo De Donno, Leon Hetzel, Yuge Ji, Ignacio L Ibarra, Sanjay R Srivatsan, Mohsen Naghypourfar, Riza M Daza, Beth Martin, et al. Predicting cellular responses to complex perturbations in high-throughput screens. *Molecular Systems Biology*, page e11517, 2023.
- [6] Michael Bereket and Theofanis Karaletsos. Modelling cellular perturbations with the sparse additive mechanism shift variational autoencoder. In *Thirty-seventh Conference on Neural Information Processing Systems*, 2023.
- [7] Yusuf Roohani, Kexin Huang, and Jure Leskovec. Predicting transcriptional outcomes of novel multigene perturbations with gears. *Nature Biotechnology*, pages 1–9, 2023.
- [8] Constantin Ahlmann-Eltze, Wolfgang Huber, and Simon Anders. Deep learning-based predictions of gene perturbation effects do not yet outperform simple linear methods. *BioRxiv*, pages 2024–09, 2024.

We thank the reviewers for their constructive and insightful comments. The following details our our point-by-point response.

Reviewer #1

One advantage of GPerturb that the authors advertised for is its interpretability, and such biological insights are going to be very important for readers of Nature Communications. What additional biological insights can researchers uncover with GPerturb?

In addition to interpretable, sparse estimates of perturbation effects, which allows the users to find subset of genes responsive to a given perturbation, our proposed method can also be used to identify interesting perturbation *patterns* by leveraging the gradient information of the estimated perturbation effects.

Denote $\hat{D}_{i,j}(x)$ the estimated perturbation effect of perturbation j on gene i at dosage level x . Due to the semiparametric specification of GPerturb, we can compute $\frac{d}{dx}\hat{D}_{i,j}(x)$, the derivative of the perturbation effects with respect to the dosage level x , exactly and efficiently (thanks to automatic differentiation) and use the derivative information to capture interesting perturbation patterns. For example, we could identify genes that are the most sensitive to the dosage of perturbation j by investigating the integral of the squared derivative $\hat{D}_i^j = \int_{A_{\min}^j}^{A_{\max}^j} \left(\frac{d}{dx}\hat{D}_{i,j}(x)\right)^2 dx$ for each gene i in the data set where A_{\min}^j, A_{\max}^j are the minimum and maximum dosage of perturbation j respectively. We choose the integral of the squared derivative as a measure of sensitivity since this quantity equals zero if and only if $\hat{D}_{i,j}(x)$ equals some constant, indicating that the perturbation effect does not depend on dosage at all, and is large only if the magnitude of rate of change in the perturbation effect is large over the interval $[A_{\min}^j, A_{\max}^j]$.

We demonstrate it using the Sciplex2 dataset: For each chemical perturbation j in {SAHA, Nutlin, Dex, BMS}, we compute the integrated squared derivative \hat{D}_i^j for all genes in Sciplex2 numerically, and report the violin plot on log scale for all genes in Figure 1B.

To illustrate the difference in sensitivity to perturbations, for each drug, we select two genes with the highest integral values of the squared derivatives, two genes with medium values (close to 1) and two genes with small values (close to 0) as examples of the most sensitive, mildly sensitive, and insensitive genes to the drug, and report their perturbed expressions in Figure 1A. We see that the genes with the highest values of the integral of the squared derivatives show a large variation with dosage compared to the mildly sensitive and insensitive genes.

Finally, for each of the four drugs in the SciPlex2 dataset, we select the top 5 most sensitive genes to the drugs. For each of the selected genes i , we report the sum of the integrals of squared derivatives $\hat{D}_i = \hat{D}_i^{\text{SAHA}} + \hat{D}_i^{\text{Nutlin}} + \hat{D}_i^{\text{Dex}} + \hat{D}_i^{\text{BMS}}$, which reflects the total sensitivity of a gene in the experiment, in Figure 1C. This illustration highlights genes, such as AKR1B10 and ALDH3A1, which are sensitive to multiple drugs and can be used to identify shared underlying biological mechanisms.

We now include this example in the main manuscript.

Figure 1: (A): Dosage vs Estimated perturbed expression. Each row corresponds to one of the four drugs (Vorinostat , Nutlin-3a, dexamethasone, BMS-345541). Left two columns consist of genes with large variation measured in integral of squared derivative, mid two columns correspond to genes showing medium variation, and the right two columns show genes with low variation. Dots corresponds to the observed expressions under different gene-perturbation pairs averaged over the test set. Shaded regions are the corresponding 95% credible band given by GPerturb. (B): Violin plot of the log of integrals of squared derivatives for all 5,000 genes in the Sciplex2 dataset. Different colours correspond to different drugs. (C): Bar plots of the sum of integrals of squared derivatives of the top 5 genes most sensitive to each of the four drugs. Note that AKR1B10 and ALDH3A1 are sensitive to more than one drugs.

Predictive performance cannot be the only evaluation metric as the model could simply be predicting well by overfitting to the noises inherent to each dataset / experiment. And the authors do not have any evaluation on the "interpretability" part of the model. How do we know if the model is interpreting true biological signal rather than interpreting random noises in the data?

We agree with the reviewer that showing good predictive performance on validation or test sets only demonstrates that the model is capable of accurately predicting the perturbed expression of samples unseen during the model's training phase (i.e. good generalisation performance), but does not provide any guarantee on the biological relevance of the model's outputs.

It is common in this area to demonstrate potential biological relevance through by gathering anecdotal evidence through post-analysis of predictions.

For instance, we could compare our model's outputs with known effects of perturbations. For example, in Sciplex2 dataset, the average posterior inclusion probability of Nutlin-3 on gene MDM2 over all 7 dosage levels is greater than 0.95, indicating presence of perturbation effect of Nutlin-3 on MDM2. This agrees with the known fact that Nutlin-3 inhibits the interaction between MDM2 and tumour suppressor p53². Similarly, the average posterior inclusion probability of SAHA (Vorinostat) on gene HDAC5, HDAC8, HDAC9 over all 7 dosage levels are all greater than 0.9, again suggesting presence of perturbation effects. This also agrees with the known biological effect of SAHA (Vorinostat)¹.

However, such anecdotal evidence represents a fundamental limitation in retrospective analyses where systematic validation of biological relevance via purely *in silico* approaches is a challenging task and ideally new data from independent, validation experiments should be acquired. In fact, our findings regarding the sensitivity to pre-processing procedures, put greater uncertainty on such anecdotal insights.

We add some additional text addressing these limitations in the Discussion section of the manuscript.

Nonetheless, we do not believe that GPerturb is predicting well by over-fitting to the noises inherent to each data set/experiment. If GPerturb had done so, we would have consistently outperformed all other methods in the comparisons. Instead, our results (Table 1) highlight that GPerturb's predictive performance is generally competitive with other state of the art methods but the key factor is that GPerturb's functionality is more flexible allowing it to be applied to more experimental settings.

Especially the pre-processing step seem to greatly affect model's interpretability, does that mean the interpretation is inherently associated to noises from how the data is pre-processed?

The fitted model and its interpretation do depend on the pre-processing steps and their implication on the noises. For example, when modelling the log-transformed data using a Gaussian likelihood, we assume the observed log-transformed count $\log(Y + 1)$ is generated by adding independent Gaussian noise ϵ to some mean value μ , i.e. $\log(Y + 1) = \mu + \epsilon$. This is equivalent to assuming $Y = \exp(\mu) \exp(\epsilon) - 1$, i.e. the observed raw count Y depends on the independent Gaussian noise ϵ in a multiplicative fashion, as $\exp(\epsilon)$ can be viewed as a independent random scaling factor. On the other hand, if we model Y directly using a Poisson or Gamma-Poisson likelihood (i.e. $Y \sim \text{Poisson}(\mu)$ or $Y \sim \text{GammaPoisson}(\mu, \alpha)$), then the scale of noise would di-

rectly depend on the scale of the mean μ , and the noise term can not be disentangled in a fashion similar to the Gaussian case. These two modelling strategies make very different assumptions on the noises, and therefore lead to outputs with different interpretations. However, we would highlight that there is no “correct” pre-processing step and modelling assumptions as the true, physical generating process of the data is unknown. Therefore our proposed method supports both discrete and continuous data, and can handle various likelihood functions, but under a common framework so that users can try different data preprocessing steps, and cross-validate the results.

Typos in the manuscript

We corrected typos in the manuscript.

Reviewer #2

The code is clearly documented, however there are some minor problems installing and running the code.

We have updated the installation pipeline. The updated package now automatically installs the correct dependencies. We also corrected typos and links in the README file. The updated code can be found at <https://github.com/hwxing3259/GPerturb>.

The provision of code for reproducibility as well as all of the data used is good, but there appear to be more minor issues that would mean the code needs modifying to run with the package provided, or at least there should be more detailed instruction

We corrected typos and errors in the notebooks. The notebooks can now be executed correctly with the provided package and other dependencies.

References

- [1] Marks, P. A. and Breslow, R. (2007). Dimethyl sulfoxide to vorinostat: development of this histone deacetylase inhibitor as an anticancer drug. *Nature biotechnology*, 25(1):84–90.
- [2] Vassilev, L. T., Vu, B. T., Graves, B., Carvajal, D., Podlaski, F., Filipovic, Z., Kong, N., Kammlott, U., Lukacs, C., Klein, C., et al. (2004). In vivo activation of the p53 pathway by small-molecule antagonists of mdm2. *Science*, 303(5659):844–848.